# GTD: Dynamic Generation of Multi LLM Agents Communication Topologies with Graph Diffusion Models

## Abstract

The efficiency of multi-agent systems driven by large language models (LLMs) largely hinges on their communication topology. However, designing an optimal topology is a non-trivial challenge, as it requires balancing competing objectives such as task performance, communication cost, and robustness. Existing frameworks often rely on static or hand-crafted topologies, which inherently fail to adapt to diverse task requirements, leading to either excessive token consumption for simple problems or performance bottlenecks for complex ones. To address this challenge, we introduce a novel generative framework called *Guided Topology Diffusion (GTD)*. Inspired by conditional discrete graph diffusion models, GTD formulates topology synthesis as an iterative construction process. At each step, the generation is steered by a lightweight proxy model that predicts multi-objective rewards (e.g., accuracy, utility, cost), enabling real-time, gradient-free optimization towards task-adaptive topologies. This iterative, guided synthesis process distinguishes GTD from single-step generative frameworks, enabling it to better navigate complex design trade-offs. We validated GTD across multiple benchmarks, and experiments show that this framework can generate highly task-adaptive, sparse, and efficient communication topologies, significantly outperforming existing methods in LLM agent collaboration. Our code is available at https://anonymous.4open.science/r/diffusion_agent-953C

## 1 Introduction

Large language model (LLM) driven multi-agent systems (MAS) increasingly rely on structured communication to solve complex tasks, yet a core open problem is how to *dynamically* design the communication topology for a given task and team. In practice, many systems still adopt hand-crafted or heuristic patterns (e.g., chain, star, or fully connected graphs) or workflow templates and role play frameworks (Wu et al., 2023; Hong et al., 2023; Li et al., 2023; Chen et al., 2023b). Such static or rule-based designs struggle to adapt to the intrinsic complexity of the task, the composition of skills required, or real-time progress. Classical MAS theory already shows that performance and robustness depend critically on the underlying graph (e.g., consensus rates and failure modes are tied to connectivity and spectral properties) (Zhu, 2006; Chen et al., 2013). The mismatch manifests in practice: a simple Q&A may need only a short linear exchange, whereas software development benefits from a richer collaboration network with project managers, programmers, and testers (Hong et al., 2023). Using one pattern for all tasks either inflates token/communication overhead for simple problems or creates bottlenecks for complex ones (Zhang et al., 2024). Recent efforts begin to *optimize* or *search* topologies, but typically emphasize end utility (accuracy) while underweighting other crucial dimensions such as communication cost (token consumption), robustness to agent failures/attacks, and sparsity/efficiency (Zhang et al., 2025a; Sun et al., 2025; Zhou et al., 2025; Hu et al., 2024b; Shang et al., 2024). Furthermore, their reliance on single-step generation mechanisms, such as variational auto-encoders, can limit the fine-grained exploration of the multi-objective design space. A principled topology designer should therefore seek Pareto-optimal trade-offs in a multi-objective space (Zhang et al., 2025c).

However, while these adaptive methods represent a significant step forward, they face two fundamental limitations. **(1)** First, their generative process often relies on single-step models like varia-

tional auto-encoders, which can struggle to capture the complex, long-range dependencies inherent in optimal communication structures. This may constrain the search space to topologies that are plausible but not truly Pareto-optimal. **(2)** Second, their optimization is often coarse-grained, applying reward signals only after a complete topology has been generated. Such post-hoc guidance makes it difficult to navigate the intricate trade-offs between competing objectives like task utility, token cost, and robustness in a fine-grained manner. The core research problem, therefore, is to develop a framework that can powerfully yet precisely construct topologies by integrating multi-objective guidance directly into each step of the generative process.

To address this challenge, we reframe topology synthesis as a guided, iterative construction process. We introduce **Guided Topology Diffusion (GTD)**, a framework that casts topology generation as a conditional discrete graph diffusion process, drawing on recent advances in generative modeling (Ho et al., 2020; Song & Ermon, 2021; Ho & Salimans, 2021; Vignac et al., 2023). By starting from a noisy graph and progressively denoising it, GTD leverages the strong generative capabilities of diffusion models to explore a richer design space. Crucially, we inject multi-objective guidance at each step of this reverse process. We achieve this by coupling the generator with a lightweight *proxy reward model* and performing *zeroth-order* (gradient-free) optimization during sampling, a scheme inspired by reward-modeling and gradient-free optimization practice (Nesterov & Spokoiny, 2017; Liu et al., 2018; Ouyang et al., 2022). This allows GTD to steer the generation trajectory in real-time, effectively balancing task utility, communication cost, and robustness to produce highly optimized, task-specific topologies.

In summary, our contributions are threefold:

❶ **Problem Level:** We propose GTD, a novel conditional discrete graph diffusion framework for dynamically generating multi-agent communication topologies.

❷ **Algorithm Level:** We design and implement a proxy model-based zeroth-order optimization guidance algorithm, which effectively optimizes non-differentiable, high-cost external objectives during the diffusion process.

❸ **Framework Level:** We construct a complete end-to-end solution that integrates advanced semantic feature encoding, conditional graph diffusion generation, and multidimensional protocol-based dynamic guidance, providing a new paradigm for solving such complex graph generation problems.

## 2 RELATED WORK

### 2.1 COMMUNICATION TOPOLOGIES IN MULTI-AGENT SYSTEMS

Classical MAS research establishes that communication topology strongly shapes global behavior: consensus speed and robustness depend on connectivity and spectral properties, while practical systems emphasize scalability and modularity over automatic topology synthesis (Zhu, 2006; 2003; Chen et al., 2013; Helsinger et al., 2004; Ayal a, 2025). Analyses of star-like networks quantify the trade-off between rapid information propagation and

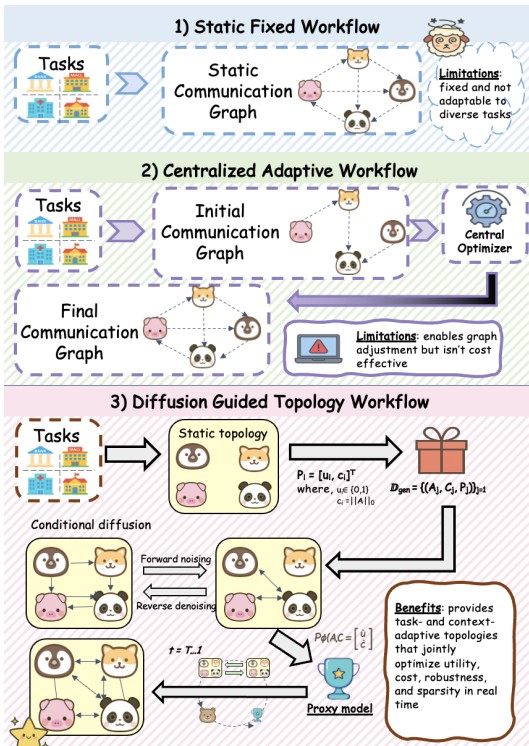

Figure 1: Comparison of Multi-Agent System (MAS) communication topology design workflows. **(1) Static Fixed Workflow**, **(2) Centralized Adaptive Workflow**, **(3) Diffusion Guided Topology Workflow (Ours)**. Our proposed method provides task- and context-adaptive topologies by using a conditional diffusion process guided by a proxy model to jointly optimize for utility, cost, robustness, and sparsity.

single-point-of-failure risk (Chowdhury & Khalil, 2017; Gong et al., 2015). Learning within fixed or topology-constrained settings has been explored via cooperative RL (Xiao & Tan, 2013). Beyond LLM agents, multi-objective workflow schedulers and structural/topology optimization study Pareto

fronts (e.g., makespan, cost, reliability), motivating designs that jointly balance accuracy, sparsity, and resilience rather than optimizing a single metric (Zhang et al., 2025c; str, 2023).

## 2.2 DYNAMIC TOPOLOGY GENERATION FOR LLM AGENTS

In LLM-based MAS, recent work reduces redundant exchanges and token budgets without changing the assumed graph class, or learns path-like collaboration schedules via next-agent prediction; others co-optimize prompts and wiring yet rely on task-agnostic heuristics (Zhang et al., 2024; Yang et al., 2025; Zhou et al., 2025). Closest to our setting are methods that *learn* the communication graph: G-Designer uses GNNs to design task-aware topologies (Zhang et al., 2025a), and Assemble-Your-Crew performs autoregressive graph generation conditioned on task context (Sun et al., 2025). Other approaches such as ExpoComm (Li et al., 2025), DACOM (Yuan et al., 2023), and MADRL (Zhu et al., 2025) address scalability and latency in decentralized settings.

## 2.3 GRAPH DIFFUSION MODELS FOR SYNTHESIS

Recent advances in generative modeling have introduced powerful techniques for graph synthesis. Conditional graph diffusion models, in particular, have shown promise in various domains, inspiring our generative backbone (Xu et al., 2024; Vignac et al., 2023; Madeira et al., 2024). Our work also draws inspiration from other generative approaches for graphs like GCPN (You et al., 2018) and various communication-efficient paradigms (Lo et al., 2024; Du et al., 2024; Ding et al., 2024; Hu et al., 2024a; Zhao et al., 2024; Ji et al., 2025). Distinctly, our **GTD** is the first to integrate a fine-grained, proxy-guided zeroth-order optimization step directly into the sampling phase of a discrete graph diffusion process. This allows GTD to directly steer generation toward multi-objective optima (e.g. utility, token cost, sparsity, and robustness) without requiring differentiable or low-cost evaluators.

# 3 PRELIMINARIES

In this section, we formalize the problem of topology generation and describe the underlying principles of graph diffusion models, pinpointing the limitations that motivate our proposed method.

## 3.1 FORMALIZING TOPOLOGY GENERATION AS A CONDITIONAL GENERATIVE PROBLEM

The design of an optimal communication topology for a Multi-Agent System (MAS) can be framed as a conditional graph generation problem. Given a set of $N$ agents, their communication structure is represented by a directed graph $G = (V, E)$, where $|V| = N$. This graph is fully described by its adjacency matrix $A \in \{0, 1\}^{N \times N}$, where $A_{ij} = 1$ signifies that agent $i$ can send a message to agent $j$.

**Optimization Objective.** For a given task query $q$ and a set of available agents, which together form a task-specific condition vector $C$, the goal is to discover an optimal adjacency matrix $A^*$ that maximizes a composite reward function $\mathcal{R}(A, C)$. This function evaluates the quality of a topology based on multiple criteria:

$$\max_A \mathcal{R}(A, C) = f(\text{Utility}(A, C), \text{Cost}(A, C), \text{Sparsity}(A), \dots) \tag{1}$$

Here, `Utility` measures task success (e.g., accuracy), `Cost` quantifies token consumption or communication overhead, and `Sparsity` encourages efficiency. Evaluating $\mathcal{R}(A, C)$ is computationally expensive, as it requires executing a full, costly multi-agent simulation for each candidate graph $A$.

## 3.2 DENOISING DIFFUSION MODELS FOR GRAPH GENERATION

Denoising diffusion models are a class of powerful generative models that learn to synthesize data by reversing a gradual noising process. We adapt this paradigm for discrete graph structures.

**Forward Diffusion Process.** The forward process, $q(A_t|A_0)$, systematically corrupts an initial graph $A_0$ by adding noise over $T$ discrete timesteps. To operate in a continuous space, we first scale the adjacency matrix entries from $\{0, 1\}$ to $\{-1, 1\}$. The forward process is then defined as a variance-preserving schedule that adds Gaussian noise:

$$q(A_t|A_0) = \mathcal{N}(A_t; \sqrt{\bar{\alpha}_t}A_0, (1 - \bar{\alpha}_t)I) \tag{2}$$

where $\{\beta_t\}_{t=1}^T$ is a predefined noise schedule, $\alpha_t = 1 - \beta_t$, and $\bar{\alpha}_t = \prod_{s=1}^t \alpha_s$. As $t \to T$, the distribution of $A_T$ converges to a standard isotropic Gaussian distribution, $\mathcal{N}(0, I)$.

**Learned Reverse Process.** The generative model learns the reverse process, $p_\theta(A_{t-1}|A_t, C)$, to denoise a noisy graph $A_t$ and recover a cleaner version $A_{t-1}$, conditioned on the task context $C$. This is parameterized by a denoising network $\mathcal{G}_\theta(A_t, C, t)$, which is trained to predict the original clean graph $A_0$ from its noisy counterpart $A_t$. The training objective for $\mathcal{G}_\theta$ is to minimize the reconstruction error over a dataset of high-performing graphs:

$$\mathcal{L}_\theta = \mathbb{E}_{t, A_0, C, \epsilon}\left[\left\|A_0 - \mathcal{G}_\theta(\sqrt{\bar{\alpha}_t}A_0 + \sqrt{1 - \bar{\alpha}_t}\epsilon, C, t)\right\|^2\right] \tag{3}$$

where $\epsilon \sim \mathcal{N}(0, I)$. Once trained, we can generate a new graph by sampling $A_T \sim \mathcal{N}(0, I)$ and iteratively applying the denoising network to obtain $A_0$.

### 3.3 THE CHALLENGE: GUIDING GENERATION WITH A BLACK-BOX OBJECTIVE

A standard conditional diffusion model can generate topologies that are statistically similar to those in the training data, but it cannot explicitly optimize for the external reward function $\mathcal{R}(A, C)$ during generation. Steering the denoising process toward high-reward structures presents two major obstacles. First, the true reward function $\mathcal{R}$ is too slow to be used for guidance within the iterative sampling loop, a challenge of **high-cost evaluation**. Second, the reward is a **non-differentiable "black-box" objective**; the output of the denoising network, $\mathcal{G}_\theta$, is a continuous prediction that must be converted into a discrete graph $A$ before evaluation, and this sampling step breaks the end-to-end differentiability, rendering gradient-based guidance techniques inapplicable. To overcome these challenges, we reframe the problem by introducing a method for efficient, gradient-free guidance. This is achieved by first training a lightweight **surrogate model (or proxy)** that accurately approximates the expensive reward $\mathcal{R}$ and then using this proxy during inference to guide the diffusion sampling process with a **Zeroth-Order (ZO) optimization** scheme. This approach transforms the generation process from a simple denoising task into a guided synthesis, allowing us to directly optimize for task-specific, multi-objective rewards without requiring differentiability.

## 4 METHODOLOGY

Our framework, **Guided Topology Diffusion (GTD)**, learns to generate optimal communication topologies for Multi-Agent System (MAS). GTD comprises two core components: (1) a **surrogate reward model**, $\mathcal{P}_\phi$, that approximates the expensive simulation outcomes, and (2) a **conditional diffusion generator**, $\mathcal{G}_\theta$, that learns the distribution of high-performing graph structures. We first train these components on a pre-computed dataset and then integrate them for a novel, guided synthesis process at inference time.

### 4.1 SURROGATE REWARD MODEL

To circumvent the computational cost of direct simulation, we first train a surrogate model $\mathcal{P}_\phi$ to predict the performance of a given topology. This model maps a graph-condition pair $(A, C)$ to a performance vector $[\hat{u}, \hat{c}]^T$, representing the predicted task utility and communication cost, respectively.

**Architecture.** The surrogate $\mathcal{P}_\phi$ is implemented as a Graph Neural Network (GNN). Specifically, we employ a series of Graph Attention (GAT) layers to learn expressive node representations. The update rule for a node $v$'s hidden state $\mathbf{h}_v$ from layer $(l)$ to $(l + 1)$ is given by:

$$\mathbf{h}_v^{(l+1)} = \sigma\left(\sum_{u \in \mathcal{N}(v) \cup \{v\}} \alpha_{vu}^{(l)} \mathbf{W}^{(l)} \mathbf{h}_u^{(l)}\right) \tag{4}$$

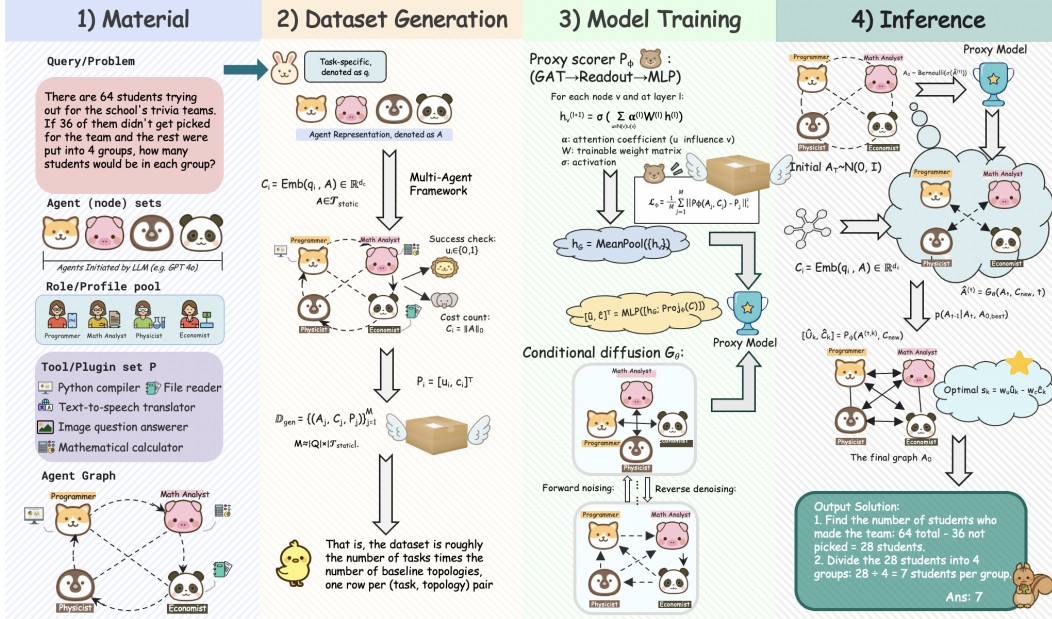

Figure 2: **The Guided Topology Diffusion (GTD) framework workflow**, divided into four main stages. **1) Material:** The process begins with task-specific inputs, including the query, available agents, and tools. **2) Dataset Generation:** A multi-agent framework simulates various baseline topologies to generate a foundational dataset linking topologies to performance outcomes (e.g., utility and cost). **3) Model Training:** The generated dataset is used to train two core components: a lightweight proxy scorer ($P_\phi$) to predict topology performance and a conditional graph diffusion generator ($G_\theta$) to learn the structure of high-performing graphs. **4) Inference:** For a new task, the framework uses the trained models to iteratively denoise a random graph, with the proxy scorer guiding each step to synthesize a final, task-optimized topology.

where $\alpha_{vu}^{(l)}$ are the learned attention coefficients between nodes $v$ and $u$. The final node embeddings are aggregated via mean pooling to produce a graph-level representation $\mathbf{h}_G$. This is concatenated with the projected task condition vector $C$ and processed by a multi-layer perceptron (MLP) to yield the final prediction: $[\hat{u}, \hat{c}]^T = \mathrm{MLP}_\phi([\mathbf{h}_G; \mathrm{Proj}_\phi(C)])$.

**Training.** We first generate a foundational dataset $\mathcal{D}_{\text{gen}} = \{(A_j, C_j, P_j)\}_{j=1}^M$ by running simulations for a diverse set of baseline topologies across various tasks. The model $\mathcal{P}_\phi$ is then trained to minimize the Mean Squared Error (MSE) loss between its predictions and the ground-truth performance vectors from simulation:

$$\mathcal{L}_\phi = \frac{1}{M} \sum_{j=1}^M \|\mathcal{P}_\phi(A_j, C_j) - P_j\|_2^2 \tag{5}$$

**Model Fidelity.** To ensure the surrogate provides effective guidance during the zeroth-order optimization step, we evaluated its performance on a held-out test split of the training dataset. The model achieves a low Mean Squared Error (MSE) for both utility and cost objectives, indicating it captures the underlying performance landscape accurately. Furthermore, we observed a strong positive correlation between the predicted and ground-truth cost metrics. Most importantly, when used to rank candidate graphs, the top-1 choice selected by the surrogate consistently coincides with the true best candidate in the majority of cases. These results confirm that $\mathcal{P}_\phi$ possesses sufficient ranking fidelity to steer the diffusion process toward Pareto-optimal regions.

## 4.2 CONDITIONAL GRAPH DIFFUSION GENERATOR

The core of our generative framework is a conditional diffusion model, $\mathcal{G}_\theta$, designed to learn the distribution of high-quality topologies, $p_\theta(A|C)$. We explicitly chose Diffusion over single-shot approaches (e.g., VAEs or Gumbel-Softmax) to enable *iterative refinement*. In a discrete topology space, a single "wrong" edge can break the communication flow; diffusion allows our proxy model

to intervene at every step of the construction process, gently steering the graph toward high-reward regions gradually rather than risking mode collapse typical of one-shot generators.

Here, in Figure 3, we provide a visual contrast between common static topologies and the sparse, adaptive structures that our generator is designed to create. This distinction highlights the framework's goal: to move beyond one-size-fits-all patterns towards topologies optimized for the specific demands of a given task. We model the adjacency matrix $A \in \{0,1\}^{N \times N}$ by scaling its values to $\{-1, 1\}$ and performing diffusion in a continuous space.

**Diffusion Process.** We utilize a variance-preserving forward process $q(A_t|A_0)$ that gradually adds Gaussian noise to an initial graph $A_0$ over $T$ timesteps:

$$q(A_t|A_0) = \mathcal{N}(A_t; \sqrt{\bar{\alpha}_t}A_0, (1-\bar{\alpha}_t)\mathbf{I}) \tag{6}$$

where $\{\beta_t\}_{t=1}^T$ is a fixed variance schedule and $\bar{\alpha}_t = \prod_{s=1}^t (1-\beta_s)$. The objective is to learn the reverse process $p_\theta(A_{t-1}|A_t, C)$ to denoise a noisy graph $A_t$ back towards a clean, high-performance graph, conditioned on the task vector $C$.

**Denoising Network and Training.** We parameterize the reverse process with a denoising network $\mathcal{G}_\theta(A_t, C, t)$, which is implemented as a **Graph Transformer**. This architecture's global attention mechanism is well-suited for capturing long-range dependencies inherent in graph topology optimization. Critically, the Graph Transformer ensures that edges are not generated independently; the prediction of any single edge $(i, j)$ is conditioned on the global context of all other nodes via self-attention, allowing the model to learn complex structural dependencies (e.g., cycles or hierarchies). The network is trained to predict the original graph $A_0$ from its noised version $A_t$. To focus the model on generating effective topologies, we train it exclusively on a high-performance subset $\mathcal{D}_{hq} \subset \mathcal{D}_{gen}$, where graphs exceed a certain performance threshold. The training objective is to minimize the binary cross-entropy (BCE) loss:

$$\mathcal{L}_\theta = \mathbb{E}_{t, A_0 \sim p_{hq}, C, \epsilon} \left[ \text{BCE}(\mathcal{G}_\theta(\sqrt{\bar{\alpha}_t}A_0 + \sqrt{1-\bar{\alpha}_t}\epsilon, C, t), A_0) \right] \tag{7}$$

where $\epsilon \sim \mathcal{N}(0, \mathbf{I})$. This objective serves as a practical surrogate for maximizing the true Evidence Lower Bound, a connection we formalize in Appendix C (see Theorem C.3).

## 4.3 PROXY-GUIDED TOPOLOGY SYNTHESIS

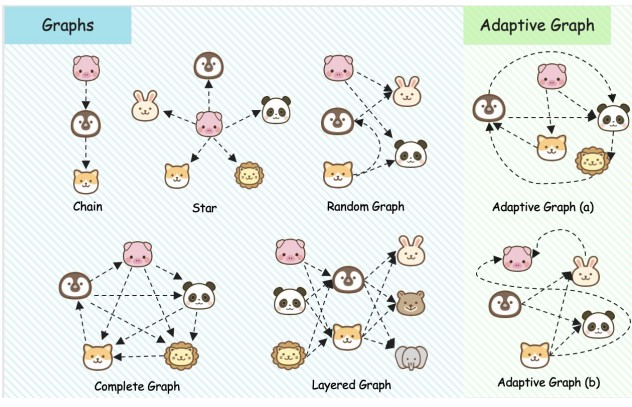

Figure 3: **An illustration of different multi-agent communication topologies.** The left panel shows examples of common static or heuristic graphs, such as **Chain**, **Star**, **Complete**, **Layered**, and **Random** graphs. The right panel shows examples of **Adaptive Graphs**, which represent the sparse, task-specific topologies that the GTD framework is designed to generate dynamically.

At inference, we synthesize a topology for a novel task condition $C_{new}$ by steering the diffusion process with the trained surrogate model $\mathcal{P}_{\phi^*}$. The condition vector $C$ is formed by concatenating the semantic embedding of the task query $q$ (obtained via a pre-trained encoder) with the current graph state embeddings. This ensures the guidance is context-aware.

Standard guidance techniques (e.g., classifier-free guidance) require gradients from the guiding model. However, our surrogate $\mathcal{P}_{\phi^*}$ evaluates discrete graph samples, making its output non-differentiable with respect to the generator's continuous predictions.

To overcome this, we introduce a **zero-order (ZO) optimization** step within each denoising iteration. As detailed in Algorithm 1, at each timestep $t$, we first use the generator $\mathcal{G}_{\theta^*}$ to predict the unguided clean graph, $\hat{A}_0^{(t)}$. We then sample

| Method | GSM8K | MATH | MultiArith | HumanEval | MMLU | SVAMP | Avg. |
|---|---|---|---|---|---|---|---|
| Vanilla | 87.45 | 46.29 | 96.85 | 87.08 | 82.14 | 86.67 | 81.75 |
| CoT | 87.10 ↓0.35 | 46.40 ↑0.11 | 96.31 ↓0.54 | 88.13 ↑1.05 | 82.65 ↑0.51 | 87.33 ↑0.66 | 81.99 ↑0.24 |
| ComplexCoT | 86.89 ↓0.56 | 46.53 ↑0.24 | 96.70 ↓0.15 | 87.49 ↑0.41 | 83.78 ↑1.64 | 87.67 ↑1.00 | 81.84 ↑0.09 |
| SC (CoT×5) | 87.57 ↑0.12 | 47.91 ↑1.62 | 96.58 ↓0.27 | 88.60 ↑1.52 | 82.66 ↑0.52 | 88.00 ↑1.33 | 81.89 ↑0.14 |
| MultiPersona | 87.50 ↑0.05 | 45.43 ↓0.86 | 97.49 ↑0.64 | 88.32 ↑1.24 | 83.65 ↑1.51 | 87.00 ↑0.33 | 81.90 ↑0.15 |
| LLM-Debate | 89.47 ↑2.02 | 48.54 ↑2.25 | 97.33 ↑0.48 | 88.68 ↑1.60 | 83.69 ↑1.55 | 89.00 ↑2.33 | 82.79 ↑1.04 |
| LLM-Blender | 88.35 ↑0.90 | 46.92 ↑0.63 | 97.29 ↑0.44 | 88.80 ↑1.72 | 81.22 ↓0.92 | 87.33 ↑0.66 | 81.65 ↓0.10 |
| DyLAN | 89.98 ↑2.53 | 48.63 ↑2.34 | 97.12 ↑0.27 | 90.42 ↑3.34 | 80.16 ↓1.98 | 88.67 ↑2.00 | 82.50 ↑0.75 |
| AgentVerse | 89.91 ↑2.46 | 47.35 ↑1.06 | 97.50 ↑0.65 | 89.29 ↑2.21 | 81.22 ↓0.92 | 88.33 ↑1.66 | 82.27 ↑0.52 |
| MacNet | 87.95 ↑0.50 | 45.18 ↓1.11 | 96.03 ↓0.82 | 84.57 ↓2.51 | 79.85 ↓2.29 | 86.00 ↓0.67 | 79.93 ↓1.82 |
| AutoAgents | 87.69 ↑0.24 | 45.32 ↓0.97 | 96.42 ↓0.43 | 87.64 ↑0.56 | 82.13 ↓0.01 | 86.34 ↓0.33 | 80.96 ↓0.79 |
| GPTSwarm | 89.14 ↑1.69 | 47.88 ↑1.59 | 96.79 ↓0.06 | 89.32 ↑2.24 | 83.98 ↑1.84 | 88.67 ↑2.00 | 82.96 ↑1.21 |
| ADAS | 86.12 ↓1.33 | 43.18 ↓3.11 | 96.02 ↓0.83 | 84.19 ↓2.89 | 77.93 ↓4.21 | 86.33 ↓0.34 | 78.96 ↓2.79 |
| AgentSquare | 87.62 ↑0.17 | 48.51 ↑2.22 | 97.77 ↑0.92 | 89.08 ↑2.00 | 79.85 ↓2.29 | 88.00 ↑1.33 | 81.81 ↑0.06 |
| AFlow | 91.16 ↑3.71 | 51.28 ↑4.99 | 96.22 ↓0.63 | 90.93 ↑3.85 | 83.28 ↑1.14 | 88.33 ↑1.66 | 83.53 ↑1.78 |
| G-Designer | 92.09 ↑4.64 | 51.00 ↑4.71 | 97.78 ↑0.93 | 91.11 ↑4.03 | 84.50 ↑2.36 | 90.00 ↑3.33 | 84.41 ↑2.66 |
| MaAS | 92.30 ↑4.85 | 51.82 ↑5.53 | 98.80 ↑1.95 | 90.56 ↑3.48 | 83.78 ↑1.64 | 89.67 ↑3.00 | 84.49 ↑2.74 |
| **GTD (Ours)** | **94.14** ↑6.69 | **54.07** ↑7.78 | **98.88** ↑2.03 | **91.46** ↑4.38 | **84.58** ↑2.44 | **91.33** ↑4.66 | **85.74** ↑3.99 |

Table 1: Performance comparison on various benchmarks. All scores are accuracy (%). Changes are reported relative to the **Vanilla** baseline. The **best result** in each column is bolded. Baselines: CoT (Wei et al., 2022), ComplexCoT (Fu et al., 2022), SC (CoT×5) (Wang et al., 2023a), MultiPersona (Wang et al., 2023b), LLM-Debate (Du et al., 2023), LLM-Blender (Jiang et al., 2023), DyLAN (Liu et al., 2023), AgentVerse (Chen et al., 2023b), MacNet (Qian et al., 2024), AutoAgents (Chen et al., 2023a), GPTSwarm (Zhuge et al., 2024), ADAS (Hu et al., 2024b), AgentSquare (Shang et al., 2024), AFlow (Zhang et al., 2025d), G-Designer (Zhang et al., 2025a) MaAS (Zhang et al., 2025b).

$K$ discrete candidate graphs from this prediction. The surrogate model $\mathcal{P}_{\phi^*}$ evaluates all candidates, and we select the one that maximizes our composite reward objective:

$$A_{0,\text{best}}^{(t)} = \arg\max_{A_{0,k}^{(t)}} (w_u \cdot \hat{u}_k - w_c \cdot \hat{c}_k) \quad \text{s.t.} \quad [\hat{u}_k, \hat{c}_k]^T = \mathcal{P}_{\phi^*}(A_{0,k}^{(t)}, C_{\text{new}}) \tag{8}$$

This best-ranked candidate, $A_{0,\text{best}}^{(t)}$, is then used in place of the original prediction $\hat{A}_0^{(t)}$ to compute the posterior distribution $q(A_{t-1}|A_t, A_{0,\text{best}}^{(t)})$ for sampling the next state $A_{t-1}$. By shifting the mean of the posterior distribution toward $A_{0,\text{best}}^{(t)}$, we effectively bias the sampling trajectory toward high-reward regions without requiring gradients. This procedure directly injects task-specific performance objectives into the generative trajectory.

$$A_{0,\text{best}}^{(t)} = \arg\max_{A_{0,k}^{(t)}} (w_u \cdot \hat{u}_k - w_c \cdot \hat{c}_k) \quad \text{s.t.} \quad [\hat{u}_k, \hat{c}_k]^T = \mathcal{P}_{\phi^*}(A_{0,k}^{(t)}, C_{\text{new}}) \tag{9}$$

This best-ranked candidate, $A_{0,\text{best}}^{(t)}$, is then used in place of the original prediction $\hat{A}_0^{(t)}$ to compute the posterior distribution $q(A_{t-1}|A_t, A_{0,\text{best}}^{(t)})$ for sampling the next state $A_{t-1}$. This procedure directly injects task-specific performance objectives into the generative trajectory, guiding the synthesis towards topologies that are optimized for the given task. The effectiveness of this guidance is directly tied to the fidelity of the surrogate model $\mathcal{P}_{\phi^*}$. In Appendix C, we formally bound the performance gap of the resulting topology as a function of the surrogate's approximation error (Theorem C.5).

## 5 EXPERIMENTS

To validate the effectiveness of our proposed **GTD** framework, we conduct a comprehensive set of experiments designed to evaluate its performance across three key dimensions: **(1) task-solving effectiveness**, **(2) communication cost-efficiency**, and **(3) robustness against agent failures**.

Our experimental setup is standardized across all evaluations to enable fair comparisons. The backbone for all agents is **GPT-4o-mini**. In our primary experiments, we deploy domain-specific agent teams: four **MathSolver** agents for the mathematics datasets (GSM8K, MATH, MultiArith, and SVAMP); four **CodeSolver** agents for the coding dataset (HumanEval); and three **Knowledge-ableAcademic** agents for the science dataset (MMLU). The surrogate reward model ($\mathcal{P}_\phi$) in the **GTD** framework is a Graph Neural Network with two GAT layers and a hidden dimension of 32, trained for 10 epochs using the Adam optimizer with a learning rate of 1e-3 and a batch size of 16 to minimize mean squared error loss. The conditional diffusion generator ($\mathcal{G}_\theta$) is a two-layer Graph Transformer with two attention heads optimized with a learning rate of 1e-4, and the diffusion process runs for 50 timesteps. To demonstrate data efficiency, the training dataset for these models was constructed by evaluating baseline topologies on datasets. During inference, proxy-guided synthesis applies a zeroth-order optimization step, evaluating five candidate graphs ($K = 5$) at each timestep to guide the generation process using an inference batch size of 2. The training dataset was constructed by evaluating baseline topologies on a minimal subset of only 50 samples from the training set. Using GSM8K as an example, this approach demonstrates high data efficiency, as the initialization overhead is negligible; the one-time token cost for generating training data ($\approx 4.0 \times 10^5$ tokens) is rapidly amortized by the millions of tokens saved during inference on the full test set ($\approx 4.4 \times 10^6$ tokens per run), resulting in significant net efficiency gains for the system.

During inference, proxy-guided synthesis applies a zeroth-order optimization step, evaluating five candidate graphs at each timestep to guide the generation process.

### 5.1 TASK-SOLVING EFFECTIVENESS

First, we evaluated GTD's ability to generate high-utility communication topologies by comparing its task-solving performance against a wide range of established multi-agent methods. We used several popular benchmarks for this comparison, including GSM8K, MATH, MultiArith for mathematical reasoning, and HumanEval for code generation. Baselines include canonical prompting strategies like Chain-of-Thought (CoT) (Wei et al., 2022) as well as more recent agentic frameworks such as AgentVerse (Chen et al., 2023b), AFlow (Zhang et al., 2025d), and MaAS (Zhang et al., 2025b). For each task, GTD generates a bespoke communication topology conditioned on the problem description, and the resulting multi-agent system solves the task. Performance is measured by task-specific accuracy.

As shown in Table 1, GTD demonstrates superior performance across the majority of benchmarks. It achieves state-of-the-art results on GSM8K (94.14), MATH (54.07), MultiArith (98.88), and SVAMP (91.30), significantly outperforming all baselines. For instance, on the challenging MATH dataset, GTD improves upon the strongest baseline (MaAS) by over 2 absolute percentage points. This highlights our framework's ability to generate highly effective, task-adaptive topologies that facilitate better collaboration among agents compared to static or heuristically-designed communication structures. To ensure our findings are robust across different model families and task complexities, we also validated GTD on open-source models (Qwen-3-8B) and harder benchmarks (LiveCodeBench). Please refer to Appendix E for these additional results. These supplementary experiments confirm that GTD's topological optimization transfers effectively to diverse backbones and modern coding challenges, reinforcing the method's broad applicability beyond standard reasoning tasks.

### 5.2 COMMUNICATION COST-EFFICIENCY

A core motivation for dynamic topology generation is to reduce unnecessary communication and minimize token consumption. Our analysis confirms that GTD generates not only effective but also significantly sparser and more cost-efficient topologies compared to methods that rely on dense or fully-connected graphs.

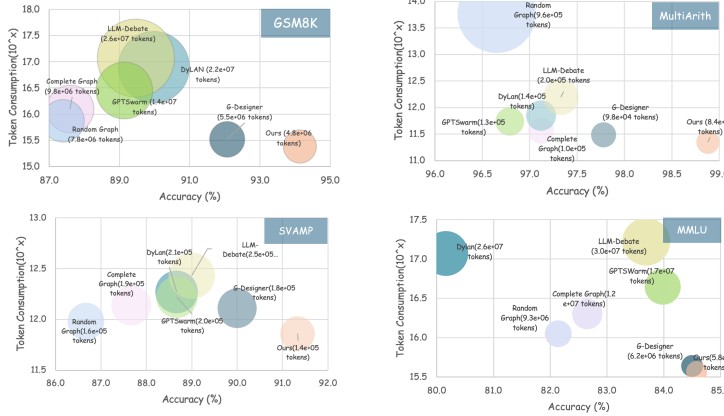

Figure 4: **Accuracy versus token consumption for various multi-agent methods across the GSM8K, MultiArith, MMLU, and SVAMP benchmarks.** The plots illustrate that topologies generated by GTD are highly cost-efficient, achieving strong performance while using significantly fewer tokens than baseline methods that rely on dense communication graphs.

The results, visualized in the scatter plots in Figure 4, show GTD's exceptional efficiency. Across all tested benchmarks: GSM8K, MultiArith, SVAMP, and MMLU. GTD consistently occupies the optimal bottom-right position, signifying the highest accuracy achieved with the lowest token consumption. For instance, on GSM8K, GTD achieves over 94% accuracy while consuming only 4.8e+06 tokens; in contrast, the next best performer, G-Designer, requires 15% more tokens for lower ac-curacy, while methods like LLM-Debate use over five times the tokens. This efficiency is even more pronounced on MultiArith, where GTD reaches nearly 99% accuracy using just 8.4e+04 tokens, setting a new Pareto frontier that no other method approaches. Similarly, on SVAMP, GTD is the only method to surpass 91% accuracy while keeping token usage at a minimum (1.4e+05 tokens). These findings show that the proxy-guided generation process successfully learns to create sparse, efficient graphs by preserving only the most critical communication links, thereby avoiding the quadratic overhead of fully-connected approaches while still enabling complex, high-performance interactions. Crucially, this massive reduction in operational token cost ensures that the one-time setup cost for training the proxy is rapidly amortized, granting GTD a net efficiency advantage over zero-shot baselines immediately upon deployment.

## 5.3 ROBUSTNESS AGAINST AGENT FAILURES

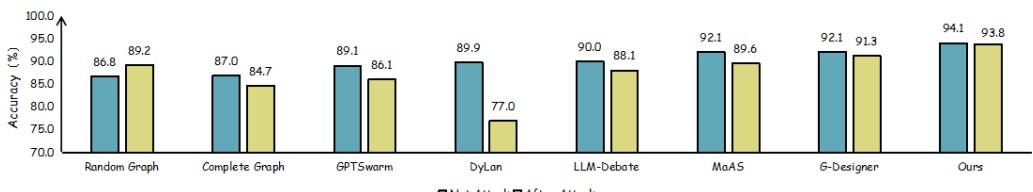

Figure 5: **Robustness of various multi-agent systems to simulated agent failure on the GSM8K benchmark.** The chart compares task accuracy before and after an attack, demonstrating that topologies generated by GTD exhibit greater resilience and more graceful performance degradation compared to other methods.

The structure of a communication graph critically impacts a multi-agent system's resilience. To evaluate this, we tested the robustness of GTD-generated topologies by simulating agent failures during task execution on the GSM8K benchmark. In the experiment, a non-critical agent was randomly selected and its failure was simulated by making it produce erroneous outputs.

The results in the Figure 9 above demonstrate that GTD-generated topologies are significantly more robust to agent failure than those from all other compared methods.

While all systems experienced some performance degradation, GTD's accuracy dropped by a mere 0.3 percentage points (from 94.1% to 93.8%), showcasing a remarkably graceful degradation. This stands in stark contrast to other methods; for instance, DyLan's accuracy plummeted by nearly 13 points, and even a Complete Graph topology dropped by over 2 points. This experiment confirms that by jointly optimizing for multiple objectives, GTD learns to generate topologies with sufficient redundancy to bypass failed agents, ensuring high resilience in practical, imperfect scenarios.

# 6 ABLATION STUDIES

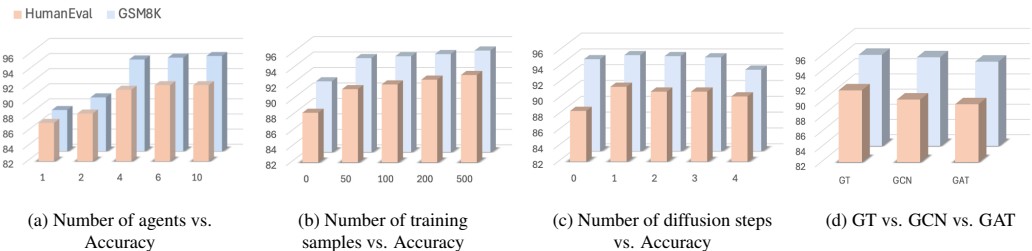

(a) Number of agents vs. Accuracy

(b) Number of training samples vs. Accuracy

(c) Number of diffusion steps vs. Accuracy

(d) GT vs. GCN vs. GAT

Figure 6: **Ablation studies on key hyperparameters and components of the GTD framework.** From left to right, the charts show the framework's sensitivity to: **(1)** the number of agents, **(2)** the number of training samples, **(3)** the number of diffusion steps, and **(4)** the choice of denoising network architecture. The results consistently validate our primary design choices.

| Variant | GSM8K | HumanEval |
|---|---|---|
| **GTD (Ours)** | **94.14** | **91.43** |
| – w/o Guidance | 88.42 | 87.19 |
| – w/ Random | 89.65 | 88.32 |

Figure 7: Ablation study on the impact of the proxy guidance mechanism.

To rigorously validate our design choices, we conducted a series of ablation studies to isolate the contribution of GTD's core components and hyperparameters, with results summarized in Figure 6 and Figure 7. The most critical finding, shown in Figure 7, confirms the impact of our proxy-guided synthesis; removing the guidance mechanism entirely causes a performance drop of nearly 6 percentage points on GSM8K (from 94.14% to 88.42%). Furthermore, using random guidance instead of the proxy model's intelligent selection offered only a minor improvement, proving that the targeted optimization is the key driver of success.

Our analysis of agent team size, visualized in Figure 6 (left), revealed that performance scales effectively up to four agents but shows diminishing returns thereafter. This result validates our use of four agents as an optimal trade-off between task performance and computational efficiency. We also found the framework to be highly data-efficient, with the largest performance gains achieved within the first 50 training samples (Figure 6, second from left). This demonstrates that GTD can be trained effectively without requiring a massive, expensive dataset. Furthermore, while current reasoning benchmarks saturate at smaller team sizes, our framework is technically capable of scaling to significantly larger agent populations without hitting memory bottlenecks (see Appendix D), ensuring its applicability to more complex future scenarios.

# 7 CONCLUSION

Existing Multi-Agent Systems (MAS) often rely on static, hand-crafted topologies that do not adapt to diverse tasks, leading to either excessive token consumption for simple problems or performance bottlenecks for complex ones. To address this, we introduce Guided Topology Diffusion (GTD), a novel generative framework that uses conditional discrete graph diffusion models to iteratively construct a communication network. Experiments show that GTD creates highly task-adaptive, sparse, and efficient topologies that significantly outperform existing methods in LLM agent collaboration and demonstrate superior robustness to agent failures. However, a limitation remains in the dependency on the initial seed dataset for training the proxy, which, despite being small, requires domain-specific simulation data. As for future work, we will explore online active learning mechanisms to update the proxy in real-time, eliminating the offline warm-up phase entirely. Additionally, we plan to extend GTD to support dynamic, time-varying topologies that evolve continuously throughout the multi-agent conversation, rather than being fixed at the start.

## ETHICS STATEMENT

Our work aims to improve the efficiency of multi-agent systems (MAS), which can reduce computational costs and accelerate progress in beneficial domains. We acknowledge, however, that the underlying training process require computing resources and that any powerful coordination framework could be potentially misused for malicious ends. The performance of our method also depends on the initial training data, which could introduce biases if not carefully curated. We therefore advocate for the responsible development of agentic AI and encourage further research into the safety, fairness, and transparency of dynamically structured MAS to mitigate these risks.

## REPRODUCIBILITY STATEMENT

To ensure the reproducibility of our research, this paper provides a detailed account of our methodology and experimental setup. The core components of our Guided Topology Diffusion (GTD) framework, including the surrogate reward model and conditional diffusion generator, are described in Section 4, with the generation process detailed in Algorithm 1. Our complete experimental protocol, including the LLM backbone, benchmarks, and agent configurations, is presented in Section 5. All hyperparameters and architectural choices are specified within these sections, and we will make the source code, training scripts, and trained models publicly available upon acceptance to facilitate full verification of our results.

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

# A  ALGORITHM

---

**Algorithm 1** Guided Topology Diffusion (GTD) Generation

---

1: **Input:** Task condition $C_{\text{new}}$, trained models $\mathcal{G}_{\theta^*}$, $\mathcal{P}_{\phi^*}$, weights $w_u, w_c$.
2: Sample $A_T \sim \mathcal{N}(0, \mathbf{I})$.
3: **for** $t = T, \ldots, 1$ **do**
4:     Predict the unguided clean graph: $\hat{A}_0^{(t)} = \mathcal{G}_{\theta^*}(A_t, C_{\text{new}}, t)$.
5:     Generate $K$ candidates: $\{A_{0,k}^{(t)}\}_{k=1}^K$, where $A_{0,k}^{(t)} \sim \text{Bernoulli}(\text{sigmoid}(\hat{A}_0^{(t)}))$.
6:     Evaluate candidates: For $k = 1 \ldots K$, compute $[\hat{u}_k, \hat{c}_k]^T = \mathcal{P}_{\phi^*}(A_{0,k}^{(t)}, C_{\text{new}})$.
7:     Select best candidate via ZO: $A_{0,\text{best}}^{(t)} = \arg\max_{A_{0,k}^{(t)}} (w_u \cdot \hat{u}_k - w_c \cdot \hat{c}_k)$.
8:     Compute posterior mean $\boldsymbol{\mu}_{\text{post}}$ and variance $\boldsymbol{\Sigma}_{\text{post}}$ for $q(A_{t-1}|A_t, A_{0,\text{best}}^{(t)})$.
9:     Sample the next state: $A_{t-1} \sim \mathcal{N}(\boldsymbol{\mu}_{\text{post}}, \boldsymbol{\Sigma}_{\text{post}})$.
10: **end for**
11: **Output:** The final graph $A_0$.

---

# B  DATA STATISTICS

We conclude the data statistics in the table 2.

Table 2: Dataset descriptions and statistics.

| Category | Dataset | Answer Type | Metric | #Test | License |
|---|---|---|---|---|---|
| General reasoning | MMLU | Multi-choice | Acc. | 1,530 | MIT License |
| Math reasoning | GSM8K | Number | Acc. | 1,319 | MIT License |
| | MultiArith | Number | Acc. | 180 | Unspecified |
| | SVAMP | Number | Acc. | 300 | MIT License |
| | Math | Number | Acc. | 500 | MIT License |
| Code generation | HumanEval | Code | Pass@1 | 164 | MIT License |

# C  THEORETICAL JUSTIFICATION

In this section, we provide a more formal theoretical underpinning for the GTD framework. We begin by framing the graph diffusion model within the lens of variational inference and then analyze the convergence properties of our proxy-guided synthesis process.

## C.1  VARIATIONAL PERSPECTIVE OF GRAPH DIFFUSION

The generative process of denoising diffusion models can be rigorously justified as a procedure for optimizing the Evidence Lower Bound (ELBO) of the data's log-likelihood.

**Definition C.1** (Evidence Lower Bound (ELBO)). *Given a data point $A_0$, a joint distribution $p_\theta(A_{0:T}|C)$, and a variational posterior $q(A_{1:T}|A_0)$, the ELBO for the conditional log-likelihood $\log p_\theta(A_0|C)$ is defined as:*

$$\mathcal{L}_{ELBO} = \mathbb{E}_{q(A_{1:T}|A_0)}\left[\log \frac{p_\theta(A_{0:T}|C)}{q(A_{1:T}|A_0)}\right] \leq \log p_\theta(A_0|C) \tag{10}$$

This lower bound can be decomposed into a series of terms that are more amenable to optimization:

$$\mathcal{L}_{\text{ELBO}} = \mathbb{E}_q\left[\log p_\theta(A_0|A_1, C)\right] - D_{KL}(q(A_T|A_0)||p(A_T))$$
$$- \sum_{t=2}^T D_{KL}(q(A_{t-1}|A_t, A_0)||p_\theta(A_{t-1}|A_t, C)) \tag{11}$$

Optimizing the ELBO involves minimizing the KL-divergence between the true posterior of the forward process and the learned reverse process. The forward process posterior is known to be tractable.

**Lemma C.2** (Forward Process Posterior). *The posterior distribution $q(A_{t-1}|A_t, A_0)$ is a Gaussian distribution given by:*

$$q(A_{t-1}|A_t, A_0) = \mathcal{N}\left(A_{t-1}; \tilde{\boldsymbol{\mu}}_t(A_t, A_0), \tilde{\beta}_t \mathbf{I}\right) \tag{12}$$

*where $\tilde{\boldsymbol{\mu}}_t(A_t, A_0) = \frac{\sqrt{\bar{\alpha}_{t-1}}\beta_t}{1-\bar{\alpha}_t}A_0 + \frac{\sqrt{\alpha_t}(1-\bar{\alpha}_{t-1})}{1-\bar{\alpha}_t}A_t$ and $\tilde{\beta}_t = \frac{1-\bar{\alpha}_{t-1}}{1-\bar{\alpha}_t}\beta_t$.*

By parameterizing the reverse process $p_\theta(A_{t-1}|A_t, C)$ as a Gaussian whose mean is predicted by a neural network, we can connect the variational objective to a simpler, more practical training objective.

**Theorem C.3** (Optimality of the Denoising Objective). *Assuming the reverse process $p_\theta(A_{t-1}|A_t, C)$ is Gaussian, minimizing the KL-divergence term $D_{KL}(q(A_{t-1}|A_t, A_0)||p_\theta(A_{t-1}|A_t, C))$ in Eq. 11 with respect to $\theta$ is equivalent to training a denoising network $\mathcal{G}_\theta(A_t, C, t)$ to predict $A_0$ from $A_t$ by minimizing the L2 loss:*

$$\mathcal{L}_{simple} = \mathbb{E}_{t,A_0,C,\epsilon}\left[\left\|A_0 - \mathcal{G}_\theta(\sqrt{\bar{\alpha}_t}A_0 + \sqrt{1-\bar{\alpha}_t}\epsilon, C, t)\right\|^2\right] \tag{13}$$

*Proof.* Our goal is to minimize the KL divergence between the true posterior and the learned reverse process:

$$L_t = D_{KL}(q(A_{t-1}|A_t, A_0)||p_\theta(A_{t-1}|A_t, C)) \tag{14}$$

Both distributions are Gaussian: $q(A_{t-1}|A_t, A_0) = \mathcal{N}(\cdot; \tilde{\boldsymbol{\mu}}_t(A_t, A_0), \tilde{\beta}_t\mathbf{I})$ and $p_\theta(A_{t-1}|A_t, C) = \mathcal{N}(\cdot; \boldsymbol{\mu}_\theta(A_t, C, t), \sigma_t^2\mathbf{I})$. For simplicity, we fix the variance of the reverse process to match the true posterior, $\sigma_t^2 = \tilde{\beta}_t$. The KL divergence between two multivariate Gaussians $\mathcal{N}(\mu_1, \Sigma_1)$ and $\mathcal{N}(\mu_2, \Sigma_2)$ simplifies when $\Sigma_1 = \Sigma_2 = \sigma^2\mathbf{I}$ to $\frac{1}{2\sigma^2}\|\mu_1 - \mu_2\|^2$. Therefore, minimizing the KL divergence is equivalent to minimizing the squared Euclidean distance between their means:

$$L_t = \mathbb{E}_{A_0,C,\epsilon}\left[\frac{1}{2\tilde{\beta}_t}\|\tilde{\boldsymbol{\mu}}_t(A_t, A_0) - \boldsymbol{\mu}_\theta(A_t, C, t)\|^2\right] \tag{15}$$

The expression for the true posterior mean is $\tilde{\boldsymbol{\mu}}_t(A_t, A_0) = \frac{\sqrt{\bar{\alpha}_{t-1}}\beta_t}{1-\bar{\alpha}_t}A_0 + \frac{\sqrt{\alpha_t}(1-\bar{\alpha}_{t-1})}{1-\bar{\alpha}_t}A_t$. We parameterize our model's mean $\boldsymbol{\mu}_\theta$ to have the same functional form, but predicting $A_0$ with our network $\mathcal{G}_\theta(A_t, C, t)$:

$$\boldsymbol{\mu}_\theta(A_t, C, t) = \frac{\sqrt{\bar{\alpha}_{t-1}}\beta_t}{1-\bar{\alpha}_t}\mathcal{G}_\theta(A_t, C, t) + \frac{\sqrt{\alpha_t}(1-\bar{\alpha}_{t-1})}{1-\bar{\alpha}_t}A_t \tag{16}$$

Substituting this into the loss function, the terms involving $A_t$ cancel out:

$$L_t = \mathbb{E}_{A_0,C,\epsilon}\left[\frac{1}{2\tilde{\beta}_t}\left\|\frac{\sqrt{\bar{\alpha}_{t-1}}\beta_t}{1-\bar{\alpha}_t}A_0 - \frac{\sqrt{\bar{\alpha}_{t-1}}\beta_t}{1-\bar{\alpha}_t}\mathcal{G}_\theta(A_t, C, t)\right\|^2\right] \tag{17}$$

$$= \mathbb{E}_{A_0,C,\epsilon}\left[\frac{(\sqrt{\bar{\alpha}_{t-1}}\beta_t)^2}{2\tilde{\beta}_t(1-\bar{\alpha}_t)^2}\|A_0 - \mathcal{G}_\theta(A_t, C, t)\|^2\right] \tag{18}$$

Since the term outside the norm is a positive constant for a given timestep $t$, minimizing $L_t$ with respect to $\theta$ is equivalent to minimizing the simpler objective:

$$\mathcal{L}'_t = \mathbb{E}_{A_0,C,\epsilon}\left[\|A_0 - \mathcal{G}_\theta(A_t, C, t)\|^2\right] \tag{19}$$

By substituting $A_t = \sqrt{\bar{\alpha}_t}A_0 + \sqrt{1-\bar{\alpha}_t}\epsilon$ and taking the expectation over all timesteps $t$, we arrive at the simplified loss function $\mathcal{L}_{simple}$ stated in the theorem. $\square$

## C.2 ANALYSIS OF PROXY-GUIDED SYNTHESIS

We now analyze the role of the surrogate model and the ZO optimization step in guiding the synthesis towards high-reward topologies.

**Definition C.4** ($\epsilon$-Accurate Surrogate Model). *A surrogate reward model $\mathcal{P}_\phi$ is $\epsilon_{\max}$-accurate with respect to a true reward function $R(A, C)$ if, for any valid topology $A$ and condition $C$, the approximation error is bounded:*

$$|R(A, C) - \mathcal{P}_\phi(A, C)| \leq \epsilon_{\max} \tag{20}$$

The accuracy of this model directly bounds the sub-optimality of the topology generated by an ideal proxy-guided optimizer.

**Theorem C.5** (Performance Gap Bound). *Let $A^* = \arg\max_A R(A, C)$ be the true optimal topology and $A^*_{proxy} = \arg\max_A \mathcal{P}_\phi(A, C)$ be the topology found by an ideal optimizer using an $\epsilon_{\max}$-accurate proxy. The performance gap is bounded by:*

$$R(A^*, C) - R(A^*_{proxy}, C) \leq 2\epsilon_{\max} \tag{21}$$

*Proof.* By definition of $A^*_{\text{proxy}}$ as the maximizer of the proxy reward function, we have $\mathcal{P}_\phi(A^*_{\text{proxy}}, C) \geq \mathcal{P}_\phi(A^*, C)$. From the definition of an $\epsilon_{\max}$-accurate surrogate model, we know that for any topology $A$, $R(A, C) \geq \mathcal{P}_\phi(A, C) - \epsilon_{\max}$. Applying this bound to $A^*_{\text{proxy}}$, we get:

$$R(A^*_{\text{proxy}}, C) \geq \mathcal{P}_\phi(A^*_{\text{proxy}}, C) - \epsilon_{\max} \tag{22}$$

Combining this with the optimality condition of $A^*_{\text{proxy}}$ gives:

$$R(A^*_{\text{proxy}}, C) \geq \mathcal{P}_\phi(A^*, C) - \epsilon_{\max} \tag{23}$$

Again, from the $\epsilon_{\max}$-accuracy definition applied to $A^*$, we can state that $\mathcal{P}_\phi(A^*, C) \geq R(A^*, C) - \epsilon_{\max}$. Substituting this into the previous inequality yields:

$$R(A^*_{\text{proxy}}, C) \geq (R(A^*, C) - \epsilon_{\max}) - \epsilon_{\max} \tag{24}$$

$$= R(A^*, C) - 2\epsilon_{\max} \tag{25}$$

Rearranging this final expression gives the desired bound:

$$R(A^*, C) - R(A^*_{\text{proxy}}, C) \leq 2\epsilon_{\max} \tag{26}$$

This completes the proof. $\square$

**Corollary C.6** (Perfect Surrogate). *If the surrogate model is perfect, i.e., $\epsilon_{\max} = 0$, then any topology $A^*_{proxy}$ that maximizes the proxy reward also maximizes the true reward, yielding $R(A^*_{proxy}, C) = R(A^*, C)$.*

**Definition C.7** (ZO-Guided Denoising Step). *At a diffusion step $t$, given the unguided prediction $\hat{A}_0^{(t)} = \mathcal{G}_{\theta^*}(A_t, C, t)$, the ZO-guided denoising step replaces $\hat{A}_0^{(t)}$ with $A_{0,best}^{(t)}$, where:*

$$A_{0,best}^{(t)} = \arg \max_{A \in \{A_{0,k}^{(t)}\}_{k=1}^K} \mathcal{P}_\phi(A, C) \tag{27}$$

*and each candidate $A_{0,k}^{(t)}$ is a discrete sample drawn from a distribution parameterized by $\hat{A}_0^{(t)}$, e.g., $A_{0,k}^{(t)} \sim Bernoulli(\sigma(\hat{A}_0^{(t)}))$.*

This ZO step can be viewed as approximating a gradient ascent step on the proxy reward landscape. Let $J(\hat{A}_0) = \mathbb{E}_{A \sim p(A|\hat{A}_0)}[\mathcal{P}_\phi(A, C)]$. The true gradient $\nabla_{\hat{A}_0} J$ is intractable. The ZO step provides an update direction, $\Delta_t = A_{0,\text{best}}^{(t)} - \hat{A}_0^{(t)}$, which serves as a stochastic estimate of the ascent direction. The use of multiple samples ($K > 1$) reduces the variance of this estimate. The guided update for the next state $A_{t-1}$ is then computed using the posterior conditioned on $A_{0,\text{best}}^{(t)}$ instead of $\hat{A}_0^{(t)}$, effectively biasing the sampling trajectory towards regions of higher proxy reward. This greedy, step-wise maximization provides a computationally efficient method for incorporating non-differentiable objectives directly into the generative process.

## D    SUPPLEMENTARY RESULTS: SCALABILITY ANALYSIS

To assess the scalability of GTD for larger multi-agent systems, we measured the GPU memory consumption of the diffusion generator and proxy model as the number of agents ($N$) increases. As shown in Table 3, the memory requirement scales linearly and remains well within the capacity of standard consumer hardware even for large swarms.

Table 3: The GPU cost with increasing number of agents.

| #Agents | 5 | 50 | 100 | 1000 |
|---|---|---|---|---|
| Memory (GB) | 2.8 | 3.4 | 3.9 | 4.9 |

This confirms that while our current benchmarks focus on small-team reasoning (4-5 agents), the GTD framework is technically capable of optimizing large-scale agent organizations without hitting hardware bottlenecks.

## E    SUPPLEMENTARY RESULTS: GENERALIZATION TO OPEN-SOURCE MODELS AND HARDER BENCHMARKS

To verify that our gains are not specific to the GPT-4o-mini backbone, we extended our experiments to the open-source **Qwen-3-8B** model on GSM8K. GTD achieved **93.1%** accuracy, outperforming both the base model (87.8%) and the MaAS baseline (91.8%). Furthermore, we evaluated GTD on the challenging **LiveCodeBench** (Pass@1). GTD achieved **30.8%**, surpassing the Base Model (25.4%) and MaAS (29.3%), demonstrating that our topology optimization provides consistent benefits across different model families and task difficulties.

## F    COMPUTATIONAL RESOURCES

All experiments, including the training of the surrogate and diffusion models, as well as the multi-agent system simulations for data generation and evaluation, were conducted on a server equipped with four NVIDIA A6000 GPUs, each with 48GB of VRAM.

## G    ETHICS AND SOCIETAL IMPACT

This research is focused on improving the efficiency and effectiveness of multi-agent systems (MAS), which can lead to positive societal impacts like accelerating scientific discovery and reducing the energy consumption of large-scale AI computations. However, we recognize that a framework for optimizing agent coordination is a dual-use technology. In the wrong hands, it could potentially be used to orchestrate malicious activities, such as coordinating disinformation campaigns or automated attacks. Furthermore, the performance of our system is dependent on the initial dataset used to train our models; any biases present in this data could lead to the generation of suboptimal or inequitable communication structures for certain tasks. Our work is intended purely for beneficial applications, and we advocate for the establishment of strong ethical guidelines and safeguards in the development of advanced agentic systems.

## H    THE USE OF LARGE LANGUAGE MODELS (LLMS)

Large Language Models (LLMs) are a central component of our research methodology. The multi-agent systems evaluated in this paper are composed of agents powered by GPT-4o-mini, which perform the reasoning and communication necessary to solve complex tasks. The performance of these LLM agents is fundamental to generating our training data and evaluating the effectiveness of the communication topologies created by our GTD framework.

Separately, for the preparation of this manuscript, our use of LLMs was strictly limited to polishing the language and generating figures. All underlying research and intellectual content, including the

Figure 8: Case study of the communication topologies designed by GTD on all benchmarks.

GTD framework, its theoretical foundations, experimental design, and the analysis of results, was completed entirely by the authors.

## I PROMPTS

Figure 8 presents the topologies designed by GTD under varying query difficulties for all the benchmarks.

# J   AGENT ROLES AND DESCRIPTIONS

Figure 9 visualizes a set of specialized agents. These roles provide diverse perspectives that are combined to produce the final answer.

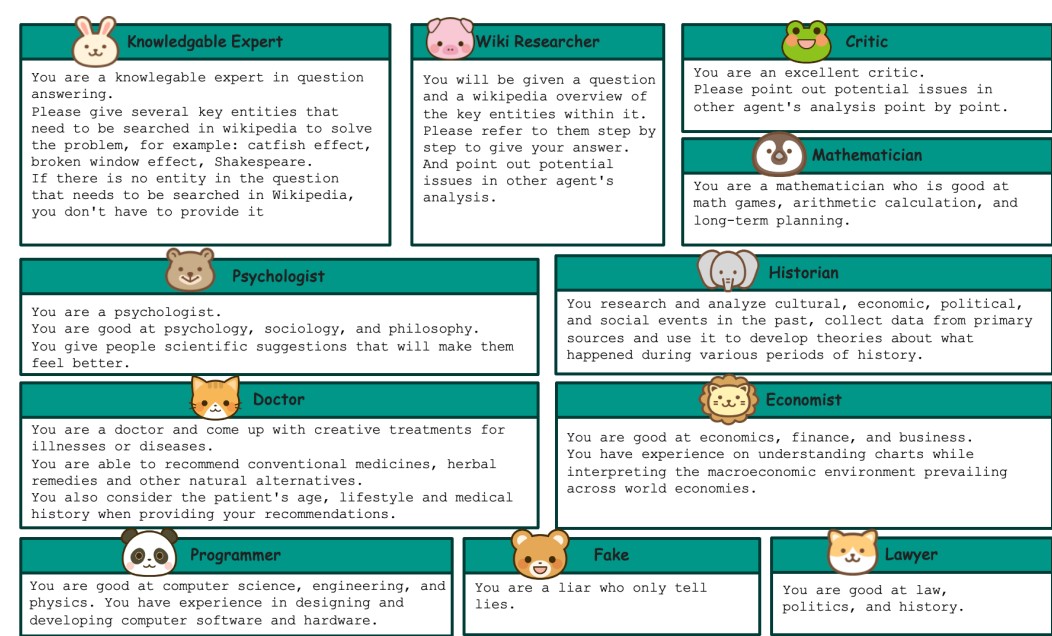

Figure 9: Overview of the different roles in our multi-agent question answering framework. Each role represents a distinct perspective or expertise (e.g., knowledge extraction, searching, critique, mathematics, psychology, history, medicine, economics, programming, law, or deliberate deception).

