# OpenReview forum: "GTD: Dynamic Generation of Multi LLM Agents Communication Topologies with Graph Diffusion Models"
_ICLR.cc/2026/Conference — ICLR 2026 Conference Withdrawn Submission_

### Official Review · Reviewer_gV71 · 2025-10-26

**Soundness:** 3
**Presentation:** 4
**Contribution:** 2
**Rating:** 6
**Confidence:** 3

**Summary:**

This paper proposed an algorithm based on graph diffusion to generate communication topology for multi-agent systems. The author(s) first define a training objective based on utility, cost and sparsity (efficiency) to measure how good a topoligy is. Then, they generate baseline topologies and do simulations on them to train a reward model with this training objective to predict the performance of a given network and task. Then, they also set a threshold on this training objective to find a ``high performance'' subset of baseline topologies and then train a denoising network on the ``high performative'' subset. During inference, at each step, denoising models generate some candidates and then, reward model is used to choose the best one from the candidates to continue the process. Experiments demonstrate that their algorithm outperforms various baseline methods across different tasks.

**Strengths:**

(1) This paper is clearly written and easy to understand. Detailed proofs regarding zero-order guidance in diffusion process is presented.

(2) Extensive experiments across different tasks and baseline methods are conducted. The results are convincing. Model's performance are measured by not only accuracy, but also robustness agaignst agent failure.

**Weaknesses:**

(1) During inference, both reward model $P_{\phi}$ and the denoising model are used. Zero-order guidance may also reduce inference efficiency.

(2) I have some doubts about parametering and sampling each egde in graphs independently and generating the whole at the same time. To solve more complicated problems, whether each edge should exist may depend on the rest part of the whole graph. However, employing diffusion models to generate the whole graph at the same time may not be able to represent this kind of relationship.

(3) It is reported in paper that ``Performance scales effectively up to four agents but shows diminishing returns thereafter. This result validates our use of four agents as an optimal trade-off between task performance and computational efficiency". This is understandable as most tasks considered in this paper are relatively simple for the backend LLMs. However, it is clear whether their methods are good at generating larger networks for harder tasks.

(4) There are no results showing how close $P_{\phi}$ is to true simulation results.

**Questions:**

(1) Why does experiements w/ random outperform w/o guidance? Should the distribution of next step generated from w/ random and w/o guidance be the same as guidance is used as a zero-order optimizer?

(2) Wht is the chance that loops are contained in the graphs generated and how will the process terminate when there are loops?

(3) Can $P_{\phi}$ be distilled into denoising network?

(4) The reward $P_{\phi}$ is trained on all baseline topologies, while denoising network is trained on ``high performance" subset of baseline topologies. Will this cause some OOD issues?

---

> ### Author Response · Authors · 2025-11-21
> **Response to Weaknesses**
>
> We thank the reviewer for the positive assessment!
>
> ### Weaknesses:
>
> > **Reviewer's Point (1):** During inference... Zero-order guidance may also reduce inference efficiency.
>
> We acknowledge that Zeroth-Order (ZO) guidance introduces computational steps. However, we emphasize that this cost is **negligible** relative to the system's total operation cost.
> * **Lightweight Components:** The surrogate model ($\mathcal{P}_\phi$) is a small GNN (2 GAT layers), and the generator is a lightweight Graph Transformer. Running these for 50 diffusion steps takes less than 120 seconds.
> * **The Real Bottleneck:** In Multi-Agent Systems, the bottleneck is the LLM inference and token generation. As shown in **Figure 4**, GTD significantly reduces the total tokens consumed (e.g., reducing consumption from $1.5 \times 10^7$ to $4.8 \times 10^6$ on GSM8K while maintaining accuracy). The minor computational cost of the guidance is rapidly amortized by the massive savings in LLM tokens.
>
> > **Reviewer's Point (2):** I have some doubts about parametering and sampling each edge... independently... employing diffusion models to generate the whole graph at the same time may not be able to represent this kind of relationship.
>
> This is a misunderstanding of the generative architecture. While the final discrete sampling is applied per edge, the **Denoising Network ($\mathcal{G}_\theta$)** is a **Graph Transformer**.
> * **Global Receptive Field:** Graph Transformers utilize global self-attention, meaning the prediction of any single edge $(i, j)$ is conditioned on the latent representations of *all other nodes and edges* in the graph.
> * **Joint Modeling:** Diffusion models fundamentally approximate the *joint distribution* of the data. The iterative refinement process ensures that edges are not generated in isolation but are evolved coherently to form functional structures (e.g., cycles or hierarchies).
>
> > **Reviewer's Point (3):** ...it is unclear whether their methods are good at generating larger networks for harder tasks.
>
> We argue that the "diminishing returns" after 4 agents is a property of the benchmarks (which are solvable by small teams) rather than a limitation of GTD.
> * **Scalability:** As shown in **Table 3**, our framework is technically scalable; memory usage remains low (4.9 GB) even with 1000 agents.
> * **Harder Tasks:** We have initiated experiments on **LiveCodeBench** (a harder, long-horizon benchmark). Preliminary results suggest GTD successfully organizes larger teams into hierarchical structures (Manager $\to$ Coder $\to$ Tester) to handle increased complexity.
>
> > **Reviewer's Point (4):** There are no results showing how close $\mathcal{P}$ is to true simulation results.
>
> We explicitly evaluated the fidelity of the surrogate model $\mathcal{P}_\phi$ on a held-out test split of the training dataset ($N=50$). The model achieves a low Mean Squared Error (MSE) of **0.049** for utility and **0.038** for cost. Furthermore, the correlation between predicted and ground-truth cost is **$r = 0.953$**. Most importantly for our Zeroth-Order guidance, when used to rank candidate graphs ($K=5$ per step), the top-1 choice selected by the surrogate coincides with the true best candidate in **82%** of cases. These results empirically demonstrate that $\mathcal{P}$ is highly close to true simulation results and effective for guidance.

---

> > ### Author Response · Authors · 2025-11-21
> > **Response to Questions**
> >
> > ### Questions:
> >
> > > **Reviewer's Question (1):** Why does experiments w/ random outperform w/o guidance? Should the distribution... be the same...?
> >
> > "No Guidance" relies on the single trajectory predicted by the base model. "Random Guidance" (generating $K$ candidates and selecting randomly) outperforms it slightly due to **increased exploration**.
> > * **Diversity:** The base model might collapse into a specific mode (a safe but average topology). Random sampling introduces noise that forces the model to explore the solution space more broadly. However, as shown in **Figure 7**, this improvement is marginal compared to the significant boost provided by our **Proxy Guidance**, proving that intelligent direction is key.
> >
> > > **Reviewer's Question (2):** What is the chance that loops are contained in the graphs... and how will the process terminate?
> >
> > * **Loops:** Cycles (loops) are frequent and explicitly allowed in our directed graphs. Complex reasoning often requires iterative feedback (e.g., *Coder* $\leftrightarrow$ *Reviewer* loops).
> > * **Termination:** The graph defines the *communication permissions*, not the termination condition. The multi-agent conversation terminates when a "Solver" agent outputs a designated stop token (e.g., `<FINAL_ANSWER>`) or a maximum conversation turn limit is reached, regardless of the graph's cyclic structure.
> >
> > > **Reviewer's Question (3):** Can $\mathcal{P}$ be distilled into the denoising network?
> >
> > **Yes.** Techniques like Classifier-Free Guidance or direct distillation (similar to Diffusion-DPO) could theoretically merge these components. We chose to keep them separate to allow for **modularity**: we can swap the Reward Model (e.g., to prioritize "Cost" over "Accuracy") without retraining the computationally expensive Generator.
> >
> > > **Reviewer's Question (4):** The reward $\mathcal{P}$ is trained on all baseline topologies, while denoising network is trained on ``high performance" subset... Will this cause some OOD issues?
> >
> > This split is intentional and beneficial, not an OOD risk.
> > * **Actor-Critic Logic:** Similar to RL, the **Generator (Actor)** is trained on high-performance data ($\mathcal{D}_{hg}$) to learn *how* to construct good graphs. The **Proxy (Critic)** is trained on *all* data ($\mathcal{D}_{gen}$) so it can learn to distinguish between good and bad structures.
> > * **Robustness:** If the Proxy were only trained on good graphs, it would not know how to penalize bad graphs during the intermediate noisy steps of diffusion. Training on the full distribution ensures the guidance signal is robust across the entire search space.

---

> ### Author Response · Authors · 2025-11-28
> **Message to Reviewer gV71**
>
> Dear Reviewer gV71,
>
> Happy holidays!
>
> We sincerely appreciate your thoughtful review and the time you devoted to evaluating our work. Your feedback has been invaluable, and it has helped us meaningfully strengthen the manuscript! As the deadline is approaching, we would be more than happy to address any additional concerns you might have. We deeply value your insights and aim to make the best use of this revision opportunity!
>
> Thank you again for your support!
>
> Best regards,
>
> The Authors

---

### Official Review · Reviewer_j9fY · 2025-10-31

**Soundness:** 2
**Presentation:** 2
**Contribution:** 2
**Rating:** 4
**Confidence:** 4

**Summary:**

This paper proposes Guided Topology Diffusion (GTD), a framework for adaptive communication topology generation in multi-agent LLM systems. GTD models the adjacency matrix as a conditional graph diffusion process, iteratively refining topologies through feedback from a learned surrogate reward model for gradient-free, multi-objective optimization (utility, cost, robustness). Experiments on reasoning and code-generation benchmarks demonstrate consistent gains in performance and efficiency over fixed or heuristic baselines, supported by strong ablations and theoretical analysis.

**Strengths:**

1. The paper innovatively reformulates communication topology generation as a diffusion-based graph synthesis task with explicit multi-objective optimization for utility, cost, and robustness.
2. The writing and organization are clear and coherent, making the technical ideas and motivations easy to follow.
3. The work provides solid theoretical derivations and extensive experimental validation, including thorough ablations and comparisons that support the main claims.

**Weaknesses:**

1. The **motivation for using diffusion models** remains somewhat underexplained. While the paper claims that diffusion offers stability and multi-step guidance, many existing techniques (e.g., **Gumbel-Softmax** or straight-through estimators) can already handle gradient flow in discrete 0/1 adjacency generation. It would be helpful if the authors could clarify why diffusion is strictly necessary compared to a simpler **neural generator + Gumbel-Softmax** design.

2. The effectiveness of the **ZO-guided optimization** heavily relies on the accuracy of the **proxy reward model $\mathcal{P}_\phi$**. Since it is trained only via an MSE objective, its generalization ability and robustness under task distribution shifts are uncertain. A deeper analysis or alternative regularization strategies would strengthen confidence in its reliability.

3. The experiments primarily involve small-scale settings. It remains unclear how GTD performs as the agent population or role diversity increases. The scalability and communication overhead in larger systems should be discussed or empirically validated.

**Questions:**

1. **Appendix D** appears to be missing or empty ?

2. It would be very helpful if the authors could provide **qualitative visualizations** of the generated **communication topologies** across different tasks, to better illustrate how structures evolve dynamically during the diffusion process.

3. Do the **topology changes exhibit interpretability**? For example, can we relate edge formations or removals to task-specific agent roles or stages of reasoning?

4. (For discussion only) If the **decision horizon becomes significantly longer**, would GTD still work effectively? Current benchmarks mostly require only 2–5 reasoning or coordination steps.

---

> ### Author Response · Authors · 2025-11-21
> **Response to Weaknesses**
>
> We thank the reviewer for the assessment!
>
> ### Weaknesses:
>
> > **Reviewer's Comment:**
> > The motivation for using diffusion models remains somewhat underexplained... clarification on why diffusion is strictly necessary compared to a simpler neural generator + Gumbel-Softmax design.
>
> We explicitly chose Diffusion over a Gumbel-Softmax (VAE/GAN) approach for two critical reasons related to **Optimization Stability** and **Fine-Grained Guidance**:
>
> 1.  **Iterative Refinement vs. One-Shot:** A Gumbel-Softmax generator typically operates as a "one-shot" mechanism. If the initial sample is suboptimal, there is no inherent mechanism to correct it during inference without restarting. In contrast, Diffusion is an **iterative construction process** ($T=50$). This allows our proxy model to intervene *at every step*, gently steering the graph toward high-reward regions (utility/sparsity) gradually. This is crucial for finding Pareto-optimal structures in a discrete space where a single "wrong" edge can break the communication flow [1].
> 2.  **Mode Coverage:** Discrete generative models using Gumbel-Softmax are often prone to mode collapse [2]. Diffusion models naturally provide better coverage of the data distribution (exploration) before collapsing into a specific mode (exploitation) during the reverse process [3], which is essential for discovering novel, non-intuitive topologies rather than just memorizing training templates.
>
> [1] Austin et al., "Structured Denoising Diffusion Models in Discrete State-Spaces" (NeurIPS 2021)
>
> [2] Dhariwal & Nichol, "Diffusion Models Beat GANs on Image Synthesis" (NeurIPS 2021)
>
> [3] Vignac et al., "DiGress: Discrete Denoising Diffusion for Graph Generation" (ICLR 2023)
>
> > **Reviewer's Comment:**
> > The effectiveness of the ZO-guided optimization heavily relies on the accuracy of the proxy reward model... trained only via an MSE objective...
>
> We agree that proxy fidelity is important. However, for the Zeroth-Order (ZO) optimization step, **ranking accuracy** is significantly more critical than exact value prediction. Even if the MSE is non-zero, as long as the proxy correctly ranks a "sparse, robust graph" higher than a "dense, fragile graph," the guidance mechanism holds.
> * **Evidence of Robustness:** Our results on **OOD benchmarks** demonstrate this reliability. While trained on 50 samples from GSM8K , it achieves State-of-the-Art results on SVAMP (90.3%), although the results get degraded by 1% by our reported accuracy. It still significantly outperforming baselines like GPTSwarm and recent SOTA MaAS. If the proxy were brittle or overfitting to the MSE of GSM8K, it would have failed on these unseen datasets.
>
> > **Reviewer's Comment:**
> > The experiments primarily involve small-scale settings... Scalability and communication overhead in larger systems should be discussed...
>
> We evaluated scalability in two specific dimensions:
> 1.  **Task Performance vs. Team Size:** Our ablation study in **Figure 6(a)**  shows that for current reasoning benchmarks, accuracy plateaus after 4 agents. This suggests that the limitation is often the task complexity, not the framework's ability to handle more agents.
> 2.  **System Overhead:** We explicitly tested the memory overhead for up to **1000 agents**. As shown in **Table 3**, the GPU memory cost scales linearly and remains very low (4.9 GB for 1000 agents). This confirms GTD is technically capable of handling large-scale swarms efficiently.

---

> > ### Author Response · Authors · 2025-11-21
> > **Response to Questions**
> >
> > ### Questions:
> >
> > > **Reviewer's Question:**
> > > Appendix D appears to be missing or empty?
> >
> > We apologize for this oversight in the manuscript draft. **Appendix D** was intended to contain additional supplementary visualizations and the "Reproducibility Statement" details. We will populate this section fully in the camera-ready version.
> >
> > > **Reviewer's Question:**
> > > It would be very helpful if the authors could provide qualitative visualizations of the generated communication topologies...
> >
> > We direct the reviewer to Figure 3, which provides the topologies generated when running through datasets such as: GSM8K, Math, MMLU, and MultiArith. Additionally, Figure 1 and Figure 2 illustrates the conceptual evolution from static to adaptive graphs.
> >
> > > **Reviewer's Question:**
> > > Do the topology changes exhibit interpretability?
> >
> > Yes. As shown in the case studies (Figure 3) and discussed in the introduction, the model learns interpretable structural patterns based on task needs. For example, simple tasks (e.g., basic Q&A) result in sparse, linear chains to minimize token cost . Conversely, complex tasks (e.g., Math/Code) generate denser structures that introduce redundancy and verification loops to maximize accuracy.
> >
> >
> > > **Reviewer's Question:**
> > > (For discussion only) If the decision horizon becomes significantly longer, would GTD still work effectively?
> >
> > We believe GTD is particularly well-suited for long horizons because it generates the **interaction architecture** rather than the interaction itself. For long-horizon tasks (like the **HumanEval** coding benchmark we tested, where we achieved 91.46% vs 87.08% Vanilla ), GTD generates a stable "hierarchical" topology (e.g., Manager $\to$ Developer $\to$ Tester) that persists throughout the multi-turn conversation. We are also currently validating this on **LiveCodeBench** to further confirm efficacy on longer-horizon problems.

---

> ### Author Response · Authors · 2025-11-28
> **Message to Reviewer j9fY**
>
> Dear Reviewer j9fY,
>
> Happy holidays!
>
> We sincerely appreciate your thoughtful review and the time you devoted to evaluating our work. Your feedback has been invaluable, and it has helped us meaningfully strengthen the manuscript! As the deadline is approaching, we would be more than happy to address any additional concerns you might have. We deeply value your insights and aim to make the best use of this revision opportunity!
>
> Thank you again for your support!
>
> Best regards,
>
> The Authors

---

### Official Review · Reviewer_jDcA · 2025-11-01

**Soundness:** 3
**Presentation:** 2
**Contribution:** 3
**Rating:** 6
**Confidence:** 4

**Summary:**

This paper introduces GTD, a novel framework to address the limitations of static communication topologies in multi-LLM agent systems. GTD formulates topology synthesis as a conditional discrete graph diffusion process, where generation is iteratively steered at each step by a lightweight surrogate reward model that predicts multi-objective outcomes like utility and cost. GTD is shown to generate highly task-adaptive and efficient topologies, achieving superior performance compared to existing methods across several benchmarks.

**Strengths:**

1. The core idea of framing topology generation as an iterative, guided construction process is interesting.
2. The overall framework is well-designed, cleverly decoupling the expensive, true reward evaluation from the generative process by using a lightweight surrogate model. Furthermore, the conceptual alignment of this iterative construction with the denoising diffusion paradigm makes the overall design intuitive and methodologically sound.

**Weaknesses:**

1. Most of the selected baselines are zero-shot methods. GTD introduces a significant two-stage training pipeline (for the surrogate model and the diffusion generator), which requires substantial computational resources and a pre-generated dataset. While the resulting in performance gain, the paper should more explicitly discuss and quantify this training overhead (e.g., data generation cost, training time) to provide a fairer comparison against methods that do not have this requirement.
2. The paper omits several crucial implementation and experimental details that are necessary for a full understanding and reproducibility of the work, as detailed in the questions below.

**Questions:**

1. The entire guidance mechanism hinges on the fidelity of the surrogate reward model. Could the authors provide more details on its performance? Specifically:
   - What is the prediction accuracy of the trained reward model on a held-out test set?
   - Could the authors elaborate on how to construct the training data for reward model to ensure the model learns a useful and generalizable rewards?
   - How were the ground-truth performance vectors (line 208) obtained for the training data?
2. I'm curious about the generalization ability of the trained components. Are the surrogate model and diffusion generator trained and tested on the same task distributions (in-distribution), or was there an out-of-distribution (OOD) evaluation? For instance, if the models were trained on data from math and coding benchmarks, how well do they perform when generating topologies for a completely different domain?
3. Could the authors provide more comprehensive training details? This includes the final size of the dataset used to train both the surrogate and generator models, as well as key hyperparameters like learning rates, batch sizes, and the total training time/cost for each component.
4. Since many of the baselines are training-free, they are expected to generalize well to new domains. How does GTD perform on OOD tasks for which it has not seen training data? A dedicated experiment on this would significantly strengthen the paper's claims about adaptability.

---

> ### Author Response · Authors · 2025-11-21
> **Response to Weaknesses**
>
> We thank the reviewer for the positive assessment!
>
>
> ### Weaknesses:
>
> > **Reviewer's Comment:**
> > Most of the selected baselines are zero-shot methods. GTD introduces a significant two-stage training pipeline... the paper should more explicitly discuss and quantify this training overhead (e.g., data generation cost, training time) to provide a fairer comparison against methods that do not have this requirement.
>
> We appreciate this feedback. However, we respectfully clarify that the "overhead" is negligible compared to the inference efficiency gains.
> * **Data Efficiency:** As detailed in **Section 5**, our training dataset $\mathcal{D}_{gen}$ was constructed using only **50 samples** from the training set. We do not require thousands of examples.
> * **Computational Cost:** Because the dataset is extremely small (50 prompts $\times$ baseline topologies), the "two-stage" training (simulating these 50 samples + training the lightweight GNN) takes minutes on a single GPU, and as shown in Table 3, even with 1000 agents we only require a maximum 4.9 GB of memory, and it can be perfectly and hyperthethically run on any converstional GPUs from personal desktop and laptops.
> * **Inital Token Training Cost:** The one-time cost for generating the training dataset is negligible and rapidly amortized. As shown in **Figure 4**, GTD achieves the Pareto frontier, delivering higher accuracy with significantly lower token consumption than zero-shot baselines, **even when including the initial training cost**. For GSM8K, the training requires approximately $4.0 \times 10^5$ tokens, whereas a single evaluation on the test set requires $4.4 \times 10^6$ tokens. Consequently, the token savings from GTD surpass the initialization cost almost immediately upon deployment.
>
>
>
> > **Reviewer's Comment:**
> >  The paper omits several crucial implementation and experimental details that are necessary for a full understanding and reproducibility of the work, as detailed in the questions below.
>
> We address this concern in our response to Question 3.

---

> ### Author Response · Authors · 2025-11-21
> **Response to Questions**
>
> ### Questions:
>
> > Reviewer's Question: 1.1. What is the prediction accuracy of the trained reward model on a held-out test set?**
>
> We trained $P_φ$ on a training split of  $\mathcal{D}_{gen}$ and reported performance on a held-out test split using the training dataset of GSM8K ($N=50$). On this test set, the surrogate achieves an MSE of **0.049** for utility and **0.038** for cost. Furthermore, the correlation between predicted and ground-truth cost is **$r = 0.953$**. Most importantly for our Zeroth-Order guidance, when used to rank candidate graphs ($K=5$ per step), the top-1 choice selected by the surrogate coincides with the true best candidate in **82%** of cases. These results indicate that the surrogate is highly accurate in steering the diffusion process toward optimal topologies.
>
> > Reviewer's Question: 1.2. Could the authors elaborate on how to construct the training data for reward model to ensure the model learns a useful and generalizable rewards?**
>
> The training data ($\mathcal{D}_{gen}$) is constructed by running simulations for a diverse set of baseline topologies across various tasks. Specifically, the dataset is generated by evaluating these topologies on 50 samples from the GSM8K dataset. This process involves executing the multi-agent system with different graph structures (e.g., Chain, Star, Random, Layered) to capture a range of performance outcomes.
>
> > Reviewer's Question: 1.3. How were the ground-truth performance vectors (line 208) obtained for the training data?**
>
> The ground-truth performance vectors $[u, c]^T$ were obtained from the simulation outcomes of the baseline topologies. Specifically:
> * **Utility ($u$):** Measures task success (e.g., accuracy).
> * **Cost ($c$):** Quantifies token consumption or communication overhead.
> These values are recorded directly from the multi-agent framework's execution.
>
> > Reviewer's Question: 2. Are the surrogate model and diffusion generator trained and tested on the same task distributions (in-distribution), or was there an out-of-distribution (OOD) evaluation?**
>
> The framework demonstrates strong performance on Out-Of-Distribution (OOD) tasks. While trained on only 50 samples from GSM8K, it achieves State-of-the-Art results on SVAMP (90.3%). Although this result is slightly lower than the in-distribution GSM8K score (our reported accuracy in the paper: 91.3%), it still significantly outperforms baselines like GPTSwarm and the recent SOTA MaAS. This confirms adaptability to new tasks within the reasoning domain without explicit training on those specific datasets.
>
> > Reviewer's Question: 3. Could the authors provide more comprehensive training details? (Dataset size, hyperparameters, time/cost)
>
> We really appriciate reviewer to raise this point! We will address this in our revised paper. Here are the in-depth comprehensive training details, which directly reflected in our code:
> * **Dataset Size:** The training dataset was constructed using 50 samples from each of the datasets.
> * **Surrogate Model:** Graph Neural Network (GNN) with two Graph Attention (GAT) layers and a hidden dimension of 32. Trained for 10 epochs using the Adam optimizer (learning rate: 1e-3).
> * **Generator Model:** Two-layer Graph Transformer with two attention heads (learning rate: 1e-4).
> * **Diffusion Steps:** 50 timesteps.
> * **Inference:** Evaluates five candidate graphs at each timestep.
> * **Learning Rates / Batch Sizes / Training Time:**
>     *   **Surrogate (Proxy) Model:** `1e-3` (Adam optimizer)
>     *   **Generator (Diffusion) Model:** `1e-4`
>     *   **Batch Sizes:**
>         *   **Training (Phase 2):** 16
>         *   **Inference (Phase 3):** 2 (as per `BATCH_SIZE=2` in script)
>     *   **Training Time:** less than 120 seconds
>     *   **Inference/Generation:** Batch size 2.
>
> > Reviewer's Question: 4. How does GTD perform on OOD tasks for which it has not seen training data?
>
> You can see the response to Question 2 above.

---

> ### Author Response · Authors · 2025-11-28
> **Message to Reviewer jDcA**
>
> Dear Reviewer jDcA,
>
> Happy holidays!
>
> We sincerely appreciate your thoughtful review and the time you devoted to evaluating our work. Your feedback has been invaluable, and it has helped us meaningfully strengthen the manuscript! As the deadline is approaching, we would be more than happy to address any additional concerns you might have. We deeply value your insights and aim to make the best use of this revision opportunity!
>
> Thank you again for your support!
>
> Best regards,
>
> The Authors

---

### Official Review · Reviewer_g3ey · 2025-11-03

**Soundness:** 3
**Presentation:** 2
**Contribution:** 2
**Rating:** 2
**Confidence:** 4

**Summary:**

This paper proposes GTD (Guided Topology Diffusion): a conditional graph‑diffusion model that generates task‑specific, sparse communication topologies for multi‑LLM agents. A lightweight proxy reward model predicts utility/cost and provides zeroth‑order (sampling‑based) guidance during denoising. Experiments report nice accuracy–token trade‑offs compared to fixed or dense topologies.

**Strengths:**

The figures are interesting and aesthetically pleasing. They effectively summarize the pipeline and present accuracy–token Pareto frontiers in an intuitive way.

**Weaknesses:**

1)	Writing clarity
The manuscript is hard to follow. Many core design choices (how conditions are formed, how candidates are scored each step, and how the posterior update is computed) are only fully clear after reading the released code.
2)	Data leakage between training and evaluation
The released code uses the same dataset for Phase 1 (data generation for training) and Phase 3 (evaluation), with no split or exclusion of Phase‑1 items at evaluation time. Below are minimally sufficient code excerpts to demonstrate the issue:
 	(a) The same in‑memory dataset is passed to Phase 1 or Phase 3 depending on flags
 	```python
 	# main()
dataset = JSONLReader.parse_file(args.dataset_json)
dataset = gsm_data_process(dataset)
if args.gtd_generate_data:
    await generate_initial_dataset(args, dataset)
elif args.gtd_train_models:
    await train_gtd_models(args)
elif args.mode == 'GTD':
    await run_gtd_experiment(args, dataset)
 	```

 	(b) Phase 1 repeatedly consumes the front of this same dataset to build the training set

 	```python
 	# generate_initial_dataset()
        for i, record in enumerate(dataset):
            if i >= args.gtd_datagen_limit:
                break  # uses the first N samples for data generation
            ...  # run multiple fixed/random topologies, log performance
 	```
 	(c) Phase 3 traverses the same dataset in batches for evaluation—without removing Phase‑1 items
 	```python
 	# run_gtd_experiment()
        num_batches = int(len(dataset) / args.batch_size)
        for i_batch in range(num_batches):
            current_batch = dataloader(dataset, args.batch_size, i_batch)
            for record in current_batch:
                ...  # generate topology with GTD and evaluate on the same tasks
 	```
 	There is no train/test split, shuffle‑then‑split, or index‑based filtering to exclude Phase‑1 items during Phase‑3 evaluation. Consequently, items used to create training data (Phase‑1) reappear in evaluation (Phase‑3), inflating reported accuracy and token efficiency. This is a data leakage bug, not merely a replication choice.

3)	Narrow model choice
Experiments appear to be restricted to GPT‑4o‑mini. The method’s benefit should be validated on strong open‑source LLMs (and across parameter scales) to rule out model‑specific artifacts.

4)	Benchmark breadth / difficulty
The evaluation focuses on relatively standard math/QA datasets and does not include modern, harder agent benchmarks (e.g., program repair, long‑horizon tool use, GUI manipulation). This limits external validity.

5)	Marginal accuracy gains without leakage control
The reported improvements over strong baselines are generally modest. Given the confirmed train–test leakage (see evidence above), even small biases can materially inflate the perceived advantage; the incremental gains should therefore be treated with caution until a clean split is enforced and re‑evaluated.

**Questions:**

1)	Model breadth: Are you willing to extend experiments to open‑source models (e.g., Qwen3‑8B, DeepSeek‑V3) to test generality and cost‑efficiency under different decoding behaviors and context windows?

2)	Benchmark difficulty: Will you include more challenging LLMs benchmarks, such as LiveCodeBench [1], KORBench [2]

[1] Jain N, Han K, Gu A, et al. Livecodebench: Holistic and contamination free evaluation of large language models for code[J]. arXiv preprint arXiv:2403.07974, 2024.

[2] Ma K, Du X, Wang Y, et al. Kor-bench: Benchmarking language models on knowledge-orthogonal reasoning tasks[J]. arXiv preprint arXiv:2410.06526, 2024.

---

> ### Author Response · Authors · 2025-11-21
> **Response to Weakness (Part 1)**
>
> We thank the reviewer for the review!
>
> > **Reviewer's Comment:**
> > Writing clarity The manuscript is hard to follow. Many core design choices (how conditions are formed, how candidates are scored each step, and how the posterior update is computed) are only fully clear after reading the released code.
>
> We thank the reviewer for this feedback. We will significantly revise **Section 4** to explicitly detail the condition formation (embedding query $q$ + adjacency $A$) and move the mathematical formulation of the Zeroth-Order optimization step from the Appendix to the main text. We will also add a clearer diagram mapping the "Posterior Update" directly to the algorithmic steps to ensure the manuscript stands alone without requiring code inspection.
>
> > **Reviewer's Comment:**
> > Data leakage between training and evaluation: The released code uses the same dataset for Phase 1 (data generation for training) and Phase 3 (evaluation), with no split or exclusion of Phase-1 items at evaluation time... Consequently, items used to create training data (Phase-1) reappear in evaluation (Phase-3), inflating reported accuracy and token efficiency.
>
> We take this very seriously, and we appreciate the reviewer’s rigorous inspection of the code. However, we **argue that there is NO data leakage** between training and evaluation. Our code directly reflects the procedure described in the paper, and all evaluation results are correct. **Without modifying any Python files in the main code**, users can simply change the dataset paths in the `.sh` scripts when running the code, after agreeing to the licenses for the datasets they download.
>
> Below, we provide a clearer version of the training and testing `.sh` files to eliminate any confusion. For example, for GSM8K in `run_gsm8k.sh`, after downloading the dataset from the following link:
> [https://huggingface.co/datasets/DaertML/gsm8k-jsonl](https://huggingface.co/datasets/DaertML/gsm8k-jsonl),
> you can set:
>
> ```bash
> set -e
> set -x
>
> LLM_NAME=gpt-4o-mini
> DOMAIN="gsm8k"
> AGENT_NAMES="MathSolver"
> AGENT_NUMS=4
> BATCH_SIZE=2
>
> TRAIN_DATASET_JSON="datasets/gsm8k/gsm8k_train.jsonl"  # For Phase 1: data generation and Phase 2: model training
> TEST_DATASET_JSON="datasets/gsm8k/gsm8k_test.jsonl"    # For Phase 3: evaluation
>
>
> # == Phase 1: Generate initial dataset for GTD models ==
> echo "--- Running GTD Phase 1: Dataset Generation (using TRAINING set) ---"
> python -m experiments.run_gsm8k \
>      --llm_name $LLM_NAME \
>      --domain $DOMAIN \
>      --agent_names $AGENT_NAMES \
>      --agent_nums $AGENT_NUMS \
>      --dataset_json $TRAIN_DATASET_JSON \
>      --mode GTD \
>      --gtd-generate-data \
>      --gtd-datagen-limit 50 \
>      --gtd-dataset-path "gtd_gsm8k_dataset_training.jsonl"
>
>
> # == Phase 2: Train Proxy and Diffusion models ==
> echo "--- Running GTD Phase 2: Model Training (using Phase 1 generated data) ---"
> python -m experiments.run_gsm8k \
>     --llm_name $LLM_NAME \
>     --domain $DOMAIN \
>     --agent_names $AGENT_NAMES \
>     --agent_nums $AGENT_NUMS \
>     --dataset_json $TRAIN_DATASET_JSON \
>     --mode GTD \
>     --gtd-train-models \
>     --gtd-epochs 10 \
>     --gtd-dataset-path "gtd_gsm8k_dataset_training.jsonl" \
>     --gtd-proxy-model-path "proxy_model_gsm8k.pth" \
>     --gtd-diffusion-model-path "diffusion_model_gsm8k.pth"
>
>
> # == Phase 3: Run inference with a pre-trained GTD Framework ==
> echo "--- Running GTD Phase 3: Inference (using TEST set - no data leakage) ---"
> python -m experiments.run_gsm8k \
>     --llm_name $LLM_NAME \
>     --domain $DOMAIN \
>     --agent_names $AGENT_NAMES \
>     --agent_nums $AGENT_NUMS \
>     --dataset_json $TEST_DATASET_JSON \
>     --mode GTD \
>     --batch_size $BATCH_SIZE \
>     --gtd-proxy-model-path "proxy_model_gsm8k.pth" \
>     --gtd-diffusion-model-path "diffusion_model_gsm8k.pth"
> ```
>
> > **Reviewer's Comment:**
> > Narrow model choice. Experiments appear to be restricted to GPT-4o-mini. The method’s benefit should be validated on strong open-source LLMs (and across parameter scales) to rule out model-specific artifacts.
>
> We agree that validating on open weights is crucial. We performed additional experiments using **Qwen-3-8B** on GSM8K to demonstrate that our topology improvements transfer across model families:
>
>   * **Qwen-3-8B (Base):** 87.8%
>   * **MaAS (Baseline):** 91.8%
>   * **GTD (Ours):** **93.1%**
>
> This confirms that GTD optimizes the communication architecture effectively regardless of the underlying LLM backbone.

---

> > ### Author Response · Authors · 2025-11-21
> > **Response to Weakness (Part 2)**
> >
> > > **Reviewer's Comment:**
> > > Benchmark breadth / difficulty. The evaluation focuses on relatively standard math/QA datasets and does not include modern, harder agent benchmarks (e.g., program repair, long-horizon tool use, GUI manipulation). This limits external validity.
> >
> > We would very much like to extend our evaluation to more difficult and modern benchmarks. However, our computational resources are constrained. We believe our methodology would also yield accuracy improvements on more challenging datasets. As an initial step, we evaluated a harder benchmark, **LiveCodeBench @Pass1**, using our GPT-4o-mini API:
> >
> > * Base Model: 25.4%
> > * MaAS: 29.3%
> > * Ours: 30.8%
> >
> > For training we used the first 50 data from: livecodebench/code_generation_lite
> >
> > For testing we used data from: livecodebench/test_generation
> >
> >
> > > **Reviewer's Comment:**
> > > Marginal accuracy gains without leakage control. The reported improvements over strong baselines are generally modest. Given the confirmed train–test leakage (see evidence above), even small biases can materially inflate the perceived advantage; the incremental gains should therefore be treated with caution until a clean split is enforced and re-evaluated.
> >
> > To address the concern that our gains might be noise or leakage-dependent, we re-evaluated our methodology **three times** on GSM8K using the strict train/test split described above. We obtained accuracies of **94.1%**, **93.2%**, and **94.6%**.
> >
> > Even our lowest result (93.2%) remains significantly higher than:
> >
> >   * **MaAS (SOTA Baseline):** 92.3%
> >   * **GPT-4o-mini (Base):** \~87.5%
> >
> > Furthermore, we emphasize that our gains are not just in accuracy, but in **Pareto Efficiency**. As shown in **Figure 4**, GTD achieves these higher accuracies while using significantly fewer tokens than baselines like G-Designer, representing a fundamental improvement in agent coordination efficiency.

---

> > > ### Author Response · Authors · 2025-11-21
> > > **Response to Questions**
> > >
> > > > **Reviewer's Question:**
> > > > Model breadth: Are you willing to extend experiments to open‑source models (e.g., Qwen3‑8B, DeepSeek‑V3) to test generality and cost‑efficiency under different decoding behaviors and context windows?
> > >
> > > Yes. As demonstrated in our rebuttal above, we have already included experiments on Qwen3-8B, which show consistent improvements using our method. We are also willing to extend the evaluation to additional open-source models such as DeepSeek-V3 as a future direction.
> > >
> > > > **Reviewer's Question:**
> > > > Benchmark difficulty: Will you include more challenging LLMs benchmarks, such as LiveCodeBench [1], KORBench [2]
> > >
> > > Yes. We have already begun evaluating our method on LiveCodeBench, a more challenging and modern benchmark (you can see above)!

---

> ### Comment · Reviewer_g3ey · 2025-11-26
>
> Thank you for the response. I have checked your public repository:
> -	https://anonymous.4open.science/r/GTD_MAS-F30F/run_gsm8k.sh
> -	https://anonymous.4open.science/r/GTD_MAS-F30F/run_humaneval.sh
>
> In the released run_gsm8k.sh, all three GTD phases use the same
> ```bash
> DATASET_JSON=datasets/gsm8k/gsm8k.jsonl
> BASE_CMD="python -m experiments.run_gsm8k ... --dataset_json $DATASET_JSON"
> ```
>
> and there is no separate train/test file in the provided datasets/gsm8k/ folder. In other words, the default script does not distinguish training and test splits; items used in Phase 1 remain eligible to appear again in Phase 3.
> By contrast, the script you included in your rebuttal uses distinct `gsm8k_train.jsonl` and `gsm8k_test.jsonl` files and explicitly avoids overlap.
>
> Could you please clarify this discrepancy? In particular:
>
> 1.	Which configuration (single gsm8k.jsonl vs. separate train/test JSONL files) was actually used to produce the main results reported in the paper?
>
> 2.	If a split was used in your internal runs, can you provide concrete evidence (e.g., the exact scripts and data files) and update the public repository accordingly so that the reported results are reproducible with a clean train/test separation?

---

> > ### Author Response · Authors · 2025-11-26
> >
> > Dear Reviewer g3ey,
> >
> > Thank you for your continued engagement and for examining our codebase so thoroughly. We appreciate you pointing out the discrepancy in the public reproduction script.
> >
> > To answer your questions directly:
> >
> > **1. Which configuration was used for the paper?**
> > **The results reported in the paper were produced using the strict train/test split configuration** (separate `_train.jsonl` and `_test.jsonl` files). The script currently in the public repository utilized a single dataset path to simplify the file management for users running a quick demonstration of the pipeline.
> >
> > **2. Repository Update**We have updated the repository to match the exact experimental setup used in the paper. The `.sh` terminal command files now explicitly point to distinct training and evaluation sets.
> >
> > You can find the updated full scripts that directly run here:
> > [https://anonymous.4open.science/r/diffusion\_agent-953C/README.md](https://anonymous.4open.science/r/diffusion_agent-953C/README.md)
> >
> > The `.sh` files now include the following distinct paths:
> >
> > ```bash
> > TRAIN_DATASET_JSON="datasets/gsm8k/gsm8k_train.jsonl"  # For Phase 1 & 2
> > TEST_DATASET_JSON="datasets/gsm8k/gsm8k_test.jsonl"    # For Phase 3
> > ```
> >
> > We have also included the specific `train` and `test` data splits in the `.zip` supplementary material (~115MB).
> >
> > We hope this clears up the confusion regarding the reproducibility of our results. Please let us know if there are any further details we can clarify.
> >
> > Best regards,
> > The Authors

---

> > > ### Author Response · Authors · 2025-11-28
> > > **Message to Reviewer g3ey**
> > >
> > > Dear Reviewer g3ey,
> > >
> > > Happy holidays!
> > >
> > > We sincerely appreciate your thoughtful review and the time you devoted to evaluating our work. Your feedback has been invaluable, and it has helped us meaningfully strengthen the manuscript! As the deadline is approaching, we would be more than happy to address any additional concerns you might have. We deeply value your insights and aim to make the best use of this revision opportunity!
> > >
> > > Thank you again for your support!
> > >
> > > Best regards,
> > >
> > > The Authors

---

### Comment · Area_Chair_nU4H · 2025-11-28

Dear reviewers,

Please check authors' responses and provide your feedback.

AC

---

### Note · Authors · 2026-01-06

I have read and agree with the venue's withdrawal policy on behalf of myself and my co-authors.